# Off-policy Policy Evaluation For Sequential Decisions Under Unobserved Confounding

**Hongseok Namkoong**[*]
Decision, Risk, and Operations Division
Columbia Business School
namkoong@gsb.columbia.edu

**Ramtin Keramati**[*]
Computational and Mathematical Engineering
Stanford University
keramati@cs.stanford.edu

**Steve Yadlowsky**[*]
Electrical Engineering
Stanford University
syadlows@stanford.edu

**Emma Brunskill**
Computer Science
Stanford University
ebrun@cs.stanford.edu

## Abstract

When observed decisions depend only on observed features, off-policy policy evaluation (OPE) methods for sequential decision problems can estimate the performance of evaluation policies before deploying them. However, this assumption is frequently violated due to unobserved confounders, unrecorded variables that impact both the decisions and their outcomes. We assess robustness of OPE methods under unobserved confounding by developing worst-case bounds on the performance of an evaluation policy. When unobserved confounders can affect every decision in an episode, we demonstrate that even small amounts of per-decision confounding can heavily bias OPE methods. Fortunately, in a number of important settings found in healthcare, policy-making, and technology, unobserved confounders may directly affect only one of the many decisions made, and influence future decisions/rewards only through the directly affected decision. Under this less pessimistic model of one-decision confounding, we propose an efficient loss-minimization-based procedure for computing worst-case bounds, and prove its statistical consistency. On simulated healthcare examples—management of sepsis and interventions for autistic children—where this is a reasonable model, we demonstrate that our method invalidates non-robust results and provides meaningful certificates of robustness, allowing reliable selection of policies under unobserved confounding.

## 1 Introduction

New technology and regulatory shifts have allowed the collection of vast amounts of data on sequential trajectories of past decisions and associated rewards, ranging from healthcare decisions and outcomes to product recommendations and purchase histories. This presents unique opportunities for using off-policy methods to inform better sequential decision-making. Leveraging prior data to evaluate the performance of a sequential decision policy (which we call the *evaluation policy*) before deploying it can reduce the need for online experimentation when doing so is expensive or risky.

A central challenge in off-policy policy evaluation (OPE) is that the estimand is inherently counterfactual: what would the rewards be *if an alternate policy had been used* (the counterfactual) instead of the behavior policy that generated the observed data (the factual). As a result, OPE requires causal reasoning about whether observed rewards were caused by observed decisions, or by a common

---

[*]Equal contribution

causal variable that simultaneously affects observed decisions and states / rewards [12]. In order to make counterfactual evaluations possible, a standard assumption—albeit often overlooked and unstated—is to require that the behavior policy does not depend on any unobserved variables that also affect the future states/rewards (no unobserved confounding). We refer to this assumption as *sequential ignorability*, following the line of works on dynamic treatment regimes [41, 36].

Sequential ignorability, however, is often violated in OPE problems where the behavior policy is unknown. In medicine, business operations, and automated systems in tech, decisions depend on unlogged features correlated with future outcomes. As an example, clinicians use visual observations or discussions with patients to inform treatment, but such information is typically not recorded; they also may rely on heuristics that are hard to quantify, and can over-extrapolate from past experience [34]. In this paper, we study a framework for quantifying the impact of unobserved confounders on OPE estimates, developing worst-case bounds on the performance of an evaluation policy. OPE estimates are often used to inform policy selection, and we are particularly interested in methods that can guide when we may be confident (or not) that an alternate decision policy should be preferred. Since OPE is generally impossible under arbitrary unobserved confounding, we begin by positing a model that explicitly limits their influence on decisions. Our proposed model is a natural extension of an influential confounding model for a single binary decision [45] to the multi-action sequential decision making setting. When unobserved confounders can affect all decisions, we illustrate in Section 3 that even small amounts of confounding can have an exponential (in the number of decisions) impact on the bias of OPE. In this sense, the accuracy of OPE can be highly unreliable under the presence of unobserved confounding that affect all decisions in multi-step horizon problems.

Fortunately, in a number of important applications, unobserved confounders may only directly affect a single decision, and influence future decisions/rewards only through this directly affected decision. As we detail shortly, in healthcare this happens when a high-level expert makes an initial decision potentially using unrecorded information, after which a standard set of protocols are followed based on recorded inputs. In financial services, this happens when humans initially screen new clients for fraud, after which decisions are made based on standard logged features. In order to evaluate new policies (e.g. fully automated systems), we need to account for unobserved confounding in the initial human-conducted screening decisions; in this scenario, the unobserved features affect subsequent decisions and rewards only through the initial decision and outcome. In other instances, it may be the case that an unobserved confounder at a particular time step is observed in the next period.

Under our less pessimistic model of single-decision confounding, we develop bounds on the expected cumulative rewards under the evaluation policy. We use functional convex duality to derive a dual relaxation, and show that it can be computed by solving a loss minimization problem. The single-decision confounding model allows us to efficiently evaluate these bounds even for continuous states, unlike the general case which requires solving an intractable nonconvex problem over likelihood ratios (which are infinite dimensional for continuous states). We prove the empirical approximation of our procedure is consistent, allowing estimation from observed past decisions.

The single-decision confounding model may not fully describe scenarios where unobserved confounders affect decisions through multiple periods. Our sensitivity analysis method is nevertheless a meaningful tool even in such scenarios, as certifying the robustness of OPE against single-decision confounding is a *necessary* condition for the conclusion of OPE to withstand multi-decision confounding. As we present in the sequel, we observe conclusions of OPE are often invalidated even under less conservative single-decision confounding, raising substantial concern for robustness of OPE under violations of sequential ignorability.

On examples of dynamic treatment regimes for autism and sepsis management, we illustrate how our single-decision confounding model allows informative bounds over meaningful amounts of confounding. Our approach provides certificates of robustness by identifying the level of unobserved confounding at which the potential bias in OPE estimates raise concerns about the validity of selecting the best policy among a set of candidates. Compared to our informative bounds, the naïve approach is prohibitively conservative and lose robustness certificates for even negligible amounts of confounding.

**Related Work** For non-sequential problems, a number of authors have proposed and studied bounds on expected rewards under unobserved confounding [33, 43, 18, 20]. In sequential settings, Tennenholtz et al. [52] recently studied settings where proxy variables of confounders satisfying structural assumptions allow identification. Zhang and Bareinboim [58] derived bounds under no restrictions on the influence of the unobserved confounder on observed decisions that they use to initialize online

learning; however, we are focused on purely offline evaluation. Much like the work of Manski [33] in the non-sequential setting, these bounds are often too conservative to guide selection of policies without performing the online learning step. Robins [42] and Brumback et al. [3] instead posit a model for how confounding in each time step affects the rewards and derive bounds under this model. Our work is complementary to these models; we do not assume access to a proxy variable, but instead assume limited influence of the unobserved confounder on the behavior policy's actions rather than the rewards, which is a natural model of confounding in many scenarios (as we describe in the sequel). Our loss minimization approach builds on work for non-sequential problems by Yadlowsky et al. [57]. Sequential decision-making problems require estimating potential outcomes that are a function of many actions, instead of one, which prevents direct application of their results. We formulate a novel loss minimization for sequential problems by deriving appropriate adjustments through time. Concurrent work by Kallus and Zhou [21] proposes bounds for policy performance at *stationarity* for infinite horizon MDPs with non adapting confounders; their setting is complementary to ours, and does not apply to non-stationary, finite-horizon problems we consider in this work (e.g. many healthcare applications are non-stationary). See Appendix A for a more detailed literature review.

**Motivating example: managing sepsis patients.** Sepsis in ICU patients accounts for one third of deaths in hospitals [13]. Difficulties of care often lead to decisions based on imperfect information, and AI-based approaches provide an opportunity for automated management of important medications for sepsis, including antibiotics and vasopressors. These approaches can decide to notify the care team when a patient should be placed on a mechanical ventilator, freeing the care team to allocate more time to critical cases. Motivated by these opportunities, and the availability of data from MIMIC-III [17], several AI-based approaches for sepsis management have been proposed [6, 25, 40].

Due to safety concerns, new policies need to be evaluated offline before online clinical validation. Confounding, however, is a serious issue in data generated from ICUs: patients in emergency departments often do not have an existing record in the hospital's electronic health system, leaving a substantial amount of patient-specific information unobserved in subsequent offline analysis. As a prominent example, comorbidities that significantly complicate the cases of sepsis [2] are often unrecorded. Private communication with an emergency department physician revealed that *initial* treatment of antibiotics at admission to the hospital are often confounded by unrecorded factors that affect the eventual outcome. For example, comorbidities such as undiagnosed heart failure can delay diagnosis of sepsis, leading to slower use of antibiotics. There is considerable discussion in the medical literature on the importance of quickly beginning antibiotic treatment, with frequently noted concerns about confounding, as these discussions are largely based on observational data [50, 51].

We consider choosing between two automated policies that differ only in initially *avoiding*, or *prescribing* antibiotics, and otherwise similarly improve current standard care. Antibiotics often are viewed as a better treatment for sepsis. In this example, unobserved factors critically effect the decision to prescribe antibiotics upon arrival; since the care team is highly trained, we assume they follow standard protocols based on recorded measurements in subsequent time steps. In Section 5, we assess the impact of confounding on OPE in this decision process with a single step of confounding.

## 2 Formulation

Notation conventions vary in the diverse set of communities interested in learning from (sequential) observational data. We use the potential outcomes [48] notation to make explicit which sequence of actions we wish to evaluate versus which sequence of actions were actually observed. In this approach, we posit all potential states and rewards exist for each possible sequence of actions, but we only observe the one corresponding to the actions taken (also known as partial, or bandit feedback), making the other potential states and rewards counterfactual. Literature in batch off policy reinforcement learning almost always assumes sequential ignorability, in which case the distribution of *potential* states and rewards are independent of the action taken by the behavior policy, conditional on the observed history. This allows us to consistently estimate counterfactuals simply based on observed outcomes. However, since our aim is to consider the impact of hypothetized confounding, clarifying the difference between the potential and observed states and rewards is cumbersome, but important.

We focus on domains modeled by episodic stochastic decision processes with a discrete set of actions. Let $\mathcal{A}_t$ be a finite set of actions available at time $t = 1, .., T$. Denote a sequence of actions $a_1 \in \mathcal{A}_1, .., a_T \in \mathcal{A}_T$ by $a_{1:T}$ (and similarly $a_{t:t'}$ for arbitrary indices $1 \leq t \leq t' \leq T$ with $a_{1:0} = \varnothing$). For

any sequence of actions $a_{1:t}$, let $S_t(a_{1:t-1})$ be the continuous-valued vector of state features, including previous rewards, $R_t(a_{1:t})$ be the reward at time $t$, and the return $Y(a_{1:T}) := \sum_{t=1}^{T} \gamma^{t-1} R_t(a_{1:t})$ be the discounted sum of rewards. We denote by $W(a_{1:T}) = (S_1, .., S_T(a_{1:T-1}), R_1(a_1), .., R_T(a_{1:T}))$ the sequence of all potential outcomes (over rewards and states) associated with the action sequence $a_{1:T}$. Any sum $\sum_{a_{1:T}}$ over action sequences is over all $a_{1:T} \in \mathcal{A}_1 \times \cdots \times \mathcal{A}_T$.

In the off-policy setting, we observe actions $A_1, .., A_T$ generated by an unknown behavior policy $\pi_1, .., \pi_T$. Let $H_t$ denote the observed history, with $H_1 := S_1$ and $H_t := (S_1, A_1, S_2(A_1), A_2, .., S_t(A_{1:t-1}))$ for $t \geq 2$. For any fixed sequence of actions $a_{1:t-1}$, denotes an instantiation of $H_t$ following the action sequence by $H_t(a_{1:t-1}) := (S_1, A_1 = a_1, S_2(a_1), .., A_{t-1} = a_{t-1}, S_t(a_{1:t-1}))$. Let $\mathcal{H}_t$ be the set of all histories. When there is *unobserved confounding* $U_t$, the behavioral policy draws actions $A_t \sim \pi_t(\cdot \mid H_t, U_t)$. Let $\pi_t(\cdot \mid H_t)$ be the conditional distribution of $A_t$ given only the observed history $H_t$, marginalizing out the unobserved confounder $U_t$.

Our goal is to bound the performance of an evaluation policy $\bar{\pi}_1, .., \bar{\pi}_T$ in a confounded sequential off-policy environment. Let $\bar{A}_t \sim \bar{\pi}_t(\cdot \mid \bar{H}_t)$ be the actions generated by the evaluation policy at time $t$, where we use $\bar{H}_t := (S_1, \bar{A}_1, S_2(\bar{A}_1), \bar{A}_2, .., S_t(\bar{A}_{1:t-1}))$ and $\bar{H}_t(a_{1:t-1}) := (S_1, \bar{A}_1 = a_1, S_2(a_1), \bar{A}_2 = a_2, .., S_t(a_{1:t-1}))$ to denote the history under the evaluation policy, analogously to the shorthands $H_t, H_t(a_{1:t-1})$; note that $\bar{H}_t$ are counterfactuals never observed in the behavioral data. We are interested in estimation of the expected cumulative reward $V^{\bar{\pi}} = \mathbb{E}[Y(\bar{A}_{1:T})]$ under the evaluation policy. Because we only observe potential outcomes $W(A_{1:t})$ evaluated at the actions $A_{1:t}$ taken by the behavior policy $\pi_t$, we need to express $\mathbb{E}[Y(\bar{A}_{1:T})]$ in terms of observable data generated by the behavioral policy $\pi_t$. To do so, we use the following definitions and assumptions.

**Assumption A.** *Overlap holds with respect to the conditional distributions over actions given only the histories between the behavior policy and the evaluation policy. That is, $\pi_t(a_t \mid H_t) > 0$ whenever $\bar{\pi}_t(a_t \mid \bar{H}_t) > 0$, for all $t$ and $a_t$, and almost every $H_t$.*

**Definition 1** (Sequential Ignorability). *A policy satisfies sequential ignorability if $\forall t$, conditional on the history generated by the policy, the action generated by the policy is independent of the potential outcomes $R_t(a_{1:t}), S_{t+1}(a_{1:t}), R_{t+1}(a_{1:t+1}), .., S_T(a_{1:T-1}), R_T(a_{1:T})$ for all $a_{1:T} \in \mathcal{A}_1 \times \cdots \mathcal{A}_T$.*

Sequential ignorability is a natural condition required for the evaluation policy to be well-defined: any additional randomization used by the evaluation policy $\bar{\pi}_t(\cdot \mid \bar{H}_t)$ cannot depend on unobserved confounders. We assume that the evaluation policy always satisfies this assumption.

**Assumption B.** *The evaluation policy satisfies sequential ignorability (Definition 1).*

OPE fundamentally requires counterfactual reasoning since we only observe states $S_t(A_{1:t-1})$ and rewards $R_t(A_{1:t})$ generated by the behavioral policy. The canonical assumption in batch off-policy RL is that sequential ignorability holds for the evaluation *and the behavior policy* [41, 42, 36]. For example, sequential ignorability allows the application of the standard importance sampling formula using the observed data; we give its proof in Section C.1. To ease notation, let $\rho_t := \frac{\bar{\pi}_t(A_t \mid \bar{H}_t(A_{1:t-1}))}{\pi_t(A_t \mid H_t)}$.

**Lemma 1.** *If sequential ignorability holds for both $\pi$ and $\bar{\pi}$, then $\mathbb{E}[Y(\bar{A}_{1:T})] = \mathbb{E}[Y(A_{1:T}) \prod_{t=1}^{T} \rho_t]$.*

## 3 Bounds under unobserved confounding

Despite the advantageous implications, it is often unrealistic to assume that the behavior policy $\pi_t$ satisfies sequential ignorability (Definition 1). If the unobserved confounder $U_t$ contains information about unseen potential rewards, then sequential ignorability doesn't hold. We now relax the sequential ignorability of the behavior policy, and instead posit a model of bounded confounding, then develop worst-case bounds on the evaluation policy performance $\mathbb{E}[Y(\bar{A}_{1:T})]$ under this model.

Without loss of generality, let the confounder $U_t$ be such that the potential outcomes are independent of $A_t$ when conditioning on $U_t$ alongside the observed states. Such an unobserved confounder always exists since we can define $U_t$ to be the tuple of all unseen potential outcomes.

**Assumption C.** *For all $t = 1, .., T$, there is a random vector $U_t$ s.t. conditional on the history $H_t$ generated by the behavior policy **and** $U_t$, $A_t \sim \pi_t(\cdot \mid H_t, U_t)$ is independent of the potential outcomes $R_t(a_{1:t}), S_{t+1}(a_{1:t}), R_{t+1}(a_{1:t+1}), .., S_T(a_{1:T-1}), R_T(a_{1:T})$ for all $a_{1:T} \in \mathcal{A}_1 \times \cdots \mathcal{A}_T$.*

Under arbitrary unobserved confounding, $\mathbb{E}[Y(\bar{A}_{1:T})]$ is not identifiable, meaning that many different values are consistent with the observed data. However, it is often plausible to posit that the unobserved

confounder $U_t$ has limited influence on the decisions of the behavior policy. In such a case, we may expect OPE estimates that (incorrectly) assume sequential ignorability may not be too biased.

Consider the following model of unobserved confounding for sequential decision making problems, which bounds confounder's influence on the behavior policy's decisions.

**Assumption D.** *For $t = 1, .., T$, there is a $\Gamma_t \geq 1$ such that for any $a_t, a'_t \in \mathcal{A}_t$*

$$\frac{\pi_t(a_t \mid H_t, U_t = u_t)\, \pi_t(a'_t \mid H_t, U_t = u'_t)}{\pi_t(a'_t \mid H_t, U_t = u_t)\, \pi_t(a_t \mid H_t, U_t = u'_t)} \leq \Gamma_t \tag{1}$$

*almost surely over $H_t$, and $u_t, u'_t$, and sequential ignorability holds conditional on $H_t$ **and** $U_t$.*

The bound (1) is a natural extension of a classical model of confounding proposed by Rosenbaum [45] for a single decision ($T = 1$) to sequential problems. For binary actions $\mathcal{A}_t = \{0, 1\}$, our bounded unobserved confounding assumption is equivalent [45] to the logistic model $\log \frac{\mathbb{P}(A_t=1|H_t,U_t)}{\mathbb{P}(A_t=0|H_t,U_t)} = \kappa(H_t) + (\log \Gamma_t)b(U_t)$ for some function $\kappa(\cdot)$ and a bounded $b(\cdot)$ taking values in $[0, 1]$.

In the sequential setting where $T > 1$, confounding can lead to exponentially large (in $T$) over-sampling of large (or small) rewards, introducing a large un-correctable bias. As an illustration, consider the simplified setting where for a single unobserved confounder $U \sim \text{Unif}(\{0, 1\})$, behavioral actions $A_1, \ldots, A_T \in \{0, 1\}$ are drawn conditionally on $U$, but independent of one another, with the conditional distribution $P(A_t = 1|U = 1) = \sqrt{\Gamma}/(1 + \sqrt{\Gamma})$ and $P(A_t = 1|U = 0) = 1/(1 + \sqrt{\Gamma})$. Let the return be $Y(a_{1:T}) = U$ for all action sequences $a_{1:T}$. Although the actions do not affect the outcome, the likelihood of observing $((A_t = 1)_{t=1}^T, Y = 1)$ is $\Gamma^{T/2}/(2(1 + \sqrt{\Gamma})^T)$, whereas the likelihood of observing $((A_t = 1)_{t=1}^T, Y = 0)$ is $1/(2(1 + \sqrt{\Gamma})^T)$. Even in the limit of infinite observations, OPE will mistakenly estimate that always taking $\bar{A}_t = 1$ leads to better rewards than always taking $\bar{A}_t = 0$. The effect of confounding is salient even in this toy example where states don't exist and rewards don't depend on actions. This has important implications for off-policy policy selection or optimization, where systematic biases can lead to selection of a poorly performing policy.

# 4 Confounding in a single decision

In many important applications, it is realistic to assume there is only a single step of confounding at a known time step $t^*$. Under this assumption, we outline in this section how we obtain a computationally and statistically feasible procedure for computing a lower (or upper) bound on the value $\mathbb{E}[Y(\bar{A}_{1:T})]$ of an evaluation policy $\bar{\pi}$. Robustness of the policy value with respect to this class of confounding effects is also a necessary (although not sufficient) requirement for robustness to confounding at multiple time steps, making this proposed method a valuable starting place for evaluating the potential effects of confounding even if one believes that there may be confounding at multiple times. After introducing precisely our model of confounding, we show in Proposition 1 how the evaluation policy value can be expressed using likelihood ratios over potential outcomes that can be used to relate the potential outcomes over observed (factual) actions with counterfactual actions not taken. These likelihood ratios over potential outcomes are unobserved, but a lower bound on the evaluation policy value can be computed by minimizing over all feasible likelihood ratios that satisfy our model of bounded confounding. Towards computational tractability, we derive a dual relaxation that can be represented as a loss minimization procedure.

We define the confounding model for when there is an unobserved confounding variable $U$ that only affects the behavior policy's action at a single time $t^\star \in [T]$. For example, in looking at the impact of confounders on antibiotics in sepsis management, it is plausible to assume that while confounders may influence the first decision when the patient arrives, later treatment decisions are un-confounded.

**Assumption E.** *For all $t \neq t^\star$, conditional on the history $H_t$ generated by the behavior policy, $A_t$ is independent of the potential outcomes $R_t(a_{1:t}), S_{t+1}(a_{1:t}), R_{t+1}(a_{1:t+1}), S_{t+2}(a_{1:t+1}), .., S_T(a_{1:T-1}), R_T(a_{1:T})$ for all $a_{1:T} \in \mathcal{A}_1 \times \cdots \mathcal{A}_T$. For $t = t^\star$, there exists a random variable $U$ such that the same conditional independence holds only when conditional on the history $H_t$ **and** $U$.*

Restricting Assumption D to a single time step $t^*$, we assume $U$ has bounded influence on $A_{t^\star}$.

**Assumption F.** *There is a $\Gamma \geq 1$ such that for any $a_{t^\star}, a'_{t^\star} \in \mathcal{A}_{t^\star}$, and a.s. over $H_{t^\star}$, and $u, u'$*

$$\frac{\pi_{t^\star}(a_{t^\star} \mid H_{t^\star}, U = u)\, \pi_{t^\star}(a'_{t^\star} \mid H_{t^\star}, U = u')}{\pi_{t^\star}(a'_{t^\star} \mid H_{t^\star}, U = u)\, \pi_{t^\star}(a_{t^\star} \mid H_{t^\star}, U = u')} \leq \Gamma. \tag{2}$$

Selecting the amount of unobserved confounding $\Gamma$ is a modeling task, and the above confounding model's simplicity and interpretability makes it advantageous for modelers to argue a plausible value of $\Gamma$. As in any applied modeling problem, the amount of unobserved confounding $\Gamma$ should be chosen with expert knowledge (e.g. by consulting doctors that make behavioral decisions). In Section 5, we give application contexts where a realistic range of $\Gamma$ can be posited. One of the most interpretable ways to assess the level of robustness to confounding is via the *design sensitivity* of the analysis [46]: the value of $\Gamma$ at which the bounds on the evaluation policy's value crosses a landmark threshold (e.g. performance of behavior policy or some known safety threshold).

By directly applying the bound (2) to adjust importance weights, we obtain a simple naive lower bound on the evaluation policy performance $\mathbb{E}[Y(\bar{A}_{1:T})]$. Details are in Section D.1.

**Lemma 2.** *Let Assumptions B, E, F hold. Then, we have*

$$\mathbb{E}[Y(\bar{A}_{1:T})] \geq \mathbb{E}\left[ Y(A_{1:T}) \times \left( \Gamma \mathbb{1}\left\{Y(A_{1:T}) < 0\right\} + \Gamma^{-1}\mathbb{1}\left\{Y(A_{1:T}) > 0\right\} \right) \times \prod_{t=1}^{T} \rho_t \right]. \qquad (3)$$

This bound is often prohibitively conservative, as we illustrate in Section 5. Instead we derive a tighter bound based on a constrained convex optimization formulation over counterfactual distributions. Under Assumption F, the likelihood ratio between observed and unobserved distribution at $t^\star$ can at most vary by a factor of $\Gamma$. Recall that $W(a_{1:T})$ is the tuple of all potential outcomes associated with the actions $a_{1:T}$. The following observation is due to Yadlowsky et al. [57, Lemma 2.1].

**Lemma 3.** *Under Assumptions E, F, for all $a_{t^\star} \neq a'_{t^\star}$, the likelihood ratio $\mathcal{L}(\cdot; H_{t^\star}, a_{t^\star}, a'_{t^\star}) := \frac{dP_W(\cdot | H_{t^\star}, A_{t^\star} = a'_{t^\star})}{dP_W(\cdot | H_{t^\star}, A_{t^\star} = a_{t^\star})}$ over the tuple of potential outcomes $W := \{W(a_{1:T})\}_{a_{1:T}}$ exists, and $\mathcal{L}(w; H_{t^\star}, a_{t^\star}, a'_{t^\star}) \leq \Gamma \mathcal{L}(w'; H_{t^\star}, a_{t^\star}, a'_{t^\star})$ for $\mathbb{P}_W(\cdot | H_{t^\star}, A_{t^\star} = a_{t^\star})$-a.s. for all $w, w'$.*

We let $\mathcal{L}(\cdot; H_{t^\star}, a_{t^\star}, a_{t^\star}) \equiv 1$. Using these (unknown) likelihood ratios, we can express the value of the evaluation policy, $\mathbb{E}[Y(\bar{A}_{1:T})]$. The proof is given in Section C.2.

**Proposition 1.** *Under Assumptions B, E, F, $\mathbb{E}[Y(\bar{A}_{1:T})]$ is equal to*

$$\mathbb{E}\left[ \prod_{t=1}^{t^\star - 1} \rho_t \sum_{a_{t^\star}, a'_{t^\star}} \bar{\pi}_{t^\star}(a_{t^\star} | \bar{H}_{t^\star}(A_{1:t^\star - 1})) \pi_{t^\star}(a_{t^\star} | H_{t^\star}) \mathbb{E}\left[ \mathcal{L}(W; H_{t^\star}, a_{t^\star}, a'_{t^\star}) Y(A_{1:T}) \prod_{t=t^\star + 1}^{T} \rho_t \,\Big|\, H_{t^\star}, A_{t^\star} = a_{t^\star} \right] \right].$$

Proposition 1 implies a natural bound on the evaluation policy value $\mathbb{E}[Y(\bar{A}_{1:T})]$ under bounded unobserved confounding. Since the likelihood ratios $\mathcal{L}(\cdot; \cdot, a_{t^\star}, a'_{t^\star})$ are fundamentally unobservable, due to their counterfactual nature we take a worst-case approach over all likelihood ratios that vary by at most a factor of $\Gamma$ (per Lemma 3), and derive a bound that only depends on observable distributions. Taking the infimum over the inner expectation in the expression derived in Proposition 1, and noting that it does not depend on $a'_{t^\star}$, define

$$\eta^\star(H_{t^\star}; a_{t^\star}) := \inf_{L \in \mathfrak{L}} \mathbb{E}\left[ L(W; H_{t^\star}) Y(A_{1:T}) \prod_{t=t^\star + 1}^{T} \rho_t \,\Big|\, H_{t^\star}, A_{t^\star} = a_{t^\star} \right]$$

where $\mathfrak{L}$ is the set of measurable mappings $L : \mathcal{W} \times \mathcal{H}_{t^\star} \to \mathbb{R}_+$ satisfying $L(w; H_{t^\star}) \leq \Gamma L(w'; H_{t^\star})$ a.s. all $w, w'$, and $\mathbb{E}[L(W; H_{t^\star}) | H_{t^\star}, A_{t^\star} = a_{t^\star}] = 1$. Since the above optimization is over infinite-dimensional likelihoods, it is difficult to compute. We use functional convex duality to derive a dual relaxation that can be computed by solving a *loss minimization* problem over any well-specified model class. This allows us to compute a meaningful lower bound to $\mathbb{E}[Y(\bar{A}_{1:T})]$ even when rewards and states are continuous, by simply fitting a model using standard supervised methods. For $(s)_+ = \max(s, 0)$ and $(s)_- = -\min(s, 0)$, define the weighted squared loss $\ell_\Gamma(z) := \frac{1}{2}(\Gamma(z)_-^2 + (z)_+^2)$.

**Theorem 2.** *Let Assumptions B, E, F hold. If $\mathbb{E}[Y(A_{1:T})^2 \prod_{t=t^\star + 1}^{T} \rho_t^2 | A_{t^\star} = a_{t^\star}, H_{t^\star}] < \infty$ a.s., then $\eta^\star(H_{t^\star}; a_{t^\star})$ is lower bounded a.s. by the unique solution*

$$\kappa^\star(H_{t^\star}; a_{t^\star}) = \underset{f(H_{t^\star})}{\operatorname{argmin}} \ \mathbb{E}\left[ \frac{\mathbb{1}\left\{A_{t^\star} = a_{t^\star}\right\}}{\pi_{t^\star}(a_{t^\star} | H_{t^\star})} \cdot \ell_\Gamma\left( Y(A_{1:T}) \prod_{t=t^\star + 1}^{T} \rho_t - f(H_{t^\star}) \right) \right].$$

See Section D.2 for the proof. From Theorem 2, our final lower bound on $\mathbb{E}[Y(\bar{A}_{1:T})]$ is given by

$$\mathbb{E}\left[ \prod_{t=1}^{t^\star - 1} \rho_t \sum_{a_{t^\star}} \bar{\pi}_{t^\star}(a_{t^\star} | \bar{H}_{t^\star}(A_{1:t^\star - 1}))(1 - \pi_{t^\star}(a_{t^\star} | H_{t^\star})) \kappa^\star(H_{t^\star}; a_{t^\star}) + \pi_{t^\star}(A_{t^\star} | H_{t^\star}) Y(A_{1:T}) \prod_{t=1}^{T} \rho_t \right]. \qquad (4)$$

Our approach yields a loss minimization problem for each possible action, where the dimension of this supervised learning problem is that of the observed history $H_{t^\star}$. If confounding occurs early, the space of possible histories is small and this learning problem becomes easier. Compared to the non-sequential setting studied by Yadlowsky et al. [57], the sequential nature of our problem requires carefully adjusting for future actions, which shows up as the product of importance weights inside the loss minimization problem. In the special case when the last decision is confounded ($t^\star = T$), our loss minimization formulation reduces to the non-sequential result due to Yadlowsky et al. [57]. The optimality results from their work would then carry over to the method suggested here. However, when confounding occurs in any other decision besides the last, the relaxation from $\eta^*$ to $\kappa^*$ makes the bounds feasible to compute, yet loose.

In cases where there is low, yet sufficient, overlap, weighted importance sampling (WIS) can dramatically reduce variance, at the cost of increased bias, with respect to the usual IS estimator. While our approach uses the IS to adjust for the differences between the behavior and evaluation policy, adjusting the bound in (4) to use WIS, instead, is straightforward. Altering the importance reweighting inside the loss function for $\kappa^\star$ to be normalized, like WIS, warrants further investigation.

Going beyond the single-decision confounding model (2) appears challenging both computationally and statistically. Under the general confounding model (1), we can formulate an optimization problem similar to that in Proposition 1 over multiple likelihood ratios corresponding to each confounded decision. Due to the multiplicative structure of the likelihood ratios, this is a nonconvex optimization problem, and convex duality does not apply. It is unclear how to develop statistically and computationally tractable reformulations of this problem akin to our loss minimization procedure, and we leave it as a topic of future research.

**Consistency**   We now show that an empirical approximation to our loss minimization problem yields a consistent estimate of $\kappa^\star(\cdot)$. We require the standard overlap assumption that requires $\rho_t$ be uniformly bounded for all $t$, that is, actions cannot be too rare under the behavior policy relative to the evaluation policy. Since it is not feasible to optimize over the class of all functions $f(H_{t^\star})$, we consider a parameterization $f_\theta(H_{t^\star})$ where $\theta \in \mathbb{R}^d$. We provide provable guarantees in the simplified setting where $\theta \mapsto f_\theta$ is linear, so that the loss minimization problem is convex. That is, we assume that $f_\theta$ is represented by a finite linear combination of some arbitrary basis functions of $H_{t^\star}$. As long as the parameterization is well-specified so that $\kappa^\star(H_{t^\star}; a_{t^\star}) = f_{\theta^\star}(H_{t^\star})$ for some $\theta^\star \in \Theta$, an empirical plug-in solution converges to $\kappa^\star$ as the number of samples $n$ grows to infinity. We let $\Theta \subseteq \mathbb{R}^d$ be our model space; our theorem allows $\Theta = \mathbb{R}^d$.

In the below result, $\widehat{\pi}_t(a_t \mid H_t)$ is a consistent estimator of $\pi_t(a_t \mid H_t)$ trained on a separate i.i.d. dataset $\mathcal{D}_n$; such estimators can be trained using sample splitting and standard supervised learning methods. Define the set $S_\epsilon$ of $\epsilon$-approximate optimizers of the empirical plug-in problem $\min_{f(H_{t^\star})} \widehat{\mathbb{E}}_n \left[ \frac{\mathbb{1}\{A_{t^\star} = a_{t^\star}\}}{\widehat{\pi}_{t^\star}(a_{t^\star} | H_{t^\star})} \ell_\Gamma \left( Y(A_{1:T}) \prod_{t=t^\star+1}^{T} \widehat{\rho}_t - f(H_{t^\star}) \right) \right]$, where $\widehat{\mathbb{E}}_n$ is the empirical distribution of the separate data, and $\widehat{\rho}_t := \frac{\bar{\pi}(A_t | \bar{H}_t(A_{1:t-1}))}{\widehat{\pi}_t(A_t | H_t(A_{1:t-1}))}$. We assume we observe i.i.d. episodes, and that episodes (unit) do not effect one another, so the observed cumulative reward is the potential outcome at the observed action sequence, $Y(A_{1:T})$. See Section D.3 for the proof of the below result.

**Theorem 3.** *Let Assumptions B, E, F hold, and let there be a $C \in (1, \infty)$ s.t. $\forall t$, $\rho_t \leq C$ a.s.. Let $\theta \mapsto f_\theta$ be linear such that $f_{\theta^\star}(\cdot) = \kappa^\star(\cdot, a_{t^\star})$ for some unique $\theta^\star \in \mathbb{R}^d$. Let $\mathbb{E}|Y(A_{1:T})|^4 < \infty$, and $\mathbb{E}[|f_\theta(H_{t^\star})|^4] < \infty$ for all $\theta \in \Theta$. If for all $t$, $\widehat{\pi}_t(\cdot|\cdot) \to \pi_t(\cdot|\cdot)$ pointwise a.s., $\widehat{\rho}_t \leq 2C$, and $(2C)^{-1} \leq \widehat{\pi}_{t^\star}(a_{t^\star}|H_{t^\star}) \leq 1$ a.s., then $\liminf_{n\to\infty} dist(\theta^\star, S_{\varepsilon_n}) \xrightarrow{p} 0 \ \forall \varepsilon_n \downarrow 0$.*

Hence, under the hypothesis of Theorem 3, a plug-in estimator is consistent for the lower bound (4).

## 5   Experiments

We demonstrate how our worst-case approach can allow reliable selection of candidate policies under confounding in realistic scenarios where confounding is primarily an issue in only a single decision. Since counterfactuals are only known in simulations, we focus on simulated healthcare examples motivated by two real OPE applications: management of sepsis and developmental interventions for autistic children. The two scenarios characterize different problem regimes: the sepsis simulator

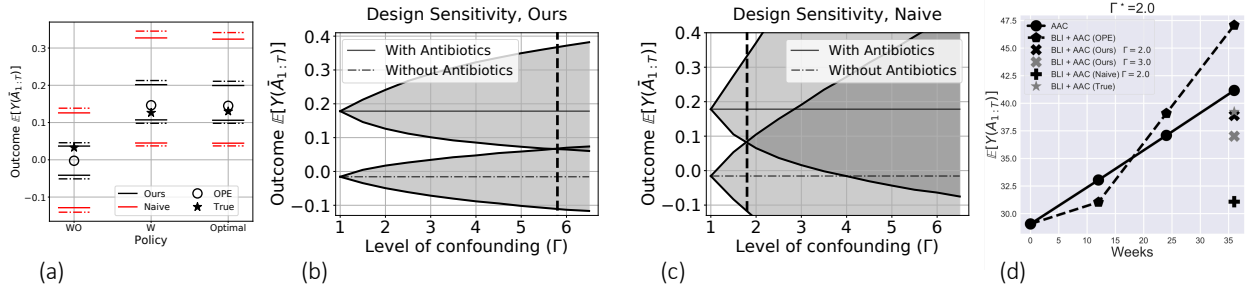

**Figure 1.** (a) Sepsis simulation with $\Gamma^{\star} = 2.0$. Dashed lines represent $95\%$ quantiles. (b, c) Sepsis design sensitivity with $\Gamma^{\star} = 5$ : (b) $\Gamma = 5.6$ for our approach, and (c) $\Gamma = 1.75$ for naïve. (d) Autism simulation. Data generation process with confounding level of $\Gamma^{\star} = 2.0$.

models discrete states, and the horizon of decision making is $T = 5$, where confounding happens at the first step; autism management simulator models continuous states and confounding happens at the last step ($T = 2 = t^{\star}$). We compare standard OPE methods that (incorrectly) assume sequential ignorability, the naïve bound from equation (3), and the bound using our proposed loss minimization approach (4); we cannot compare to the bound proposed by Zhang and Bareinboim [58] as it is prohibitively costly to compute in large or continuous state spaces, and does not scale to our settings. Our results demonstrate scalability of our loss minimization approach in both discrete and continuous settings, as well as horizons between $5 \sim 10$. Beyond 10 time steps, we observe overlap becomes a problem, and statistical estimation becomes challenging. All the code required to reproduce our experiments can be found in the Supplementary Materials.

**Managing sepsis for ICU patients** Recalling Section 1, ICU observational data on sepsis patients often lack information about important confounders, such as unrecorded comorbidities that affect a clinician's *initial decision* whether to administer antibiotics [2]. We assume the highly-trained clinical care team follows standard protocols based on vitals signs and lab measurements after the first time step, and hence their subsequent decisions are unconfounded (guaranteeing Assumption E). Using the sepsis simulator developed by Oberst and Sontag [38], we consider a scenario where automated policies have been proposed, and we wish to evaluate their benefits. We evaluate three different policies, all of which only differ in their initial prescription of antibiotics, and otherwise act optimally: *Without antibiotics (WO)*, does not administer antibiotics initially, whereas the second policy, *with antibiotics (W)*, always administers antibiotics initially; the *optimal* policy learned by policy iteration.

To simulate unrecorded comorbidities that could introduce confounding, we simulate an unobserved confounder associated with favorable state transitions. At $t = 1$, we take the optimal action with respect to all other options (vasopressors and mechanical ventilation), and administer antibiotics with probability $\sqrt{\Gamma^{\star}}/(1 + \sqrt{\Gamma^{\star}})$ if the confounding variable is large, and with probability $1/(1 + \sqrt{\Gamma^{\star}})$ if the confounding variable is small. This satisfies Assumption F with level $\Gamma^{\star}$. Since $\Gamma^{\star}$ is unknown to the procedure used to estimate (bounds on) the evaluation policy, we run our method with varying levels of $\Gamma$, and look at thresholds at which the bounds on the performance of evaluation policies cross each other (which we refer to as the *design sensitivity*). To generate observational data, we assume the behaviour policy (care team) acts nearly optimally, except for some randomness due to challenges in the ICU; this guarantees overlap. For $t \geq 2$, the behavior policy takes the optimal next treatment action with probability $0.85$, and otherwise switches the vasopressor status, independent of the confounders; we refer the reader to Section E.1 for more details.

First, assume that the true level of confounding is known, so that $\Gamma = \Gamma^{\star}$. Figure 1 (a) plots the value of the three evaluation decision policies estimated using the data generated with $\Gamma^{\star} = 2.0$, a low level of confounding. The true value of the without antibiotics (WO) policy is lower than the true value of the with antibiotics (W) policy. However, confounding leads standard OPE methods (evaluated using weighted importance sampling) that assume sequential ignorability for the behavior policy to underestimate the performance of the WO policy, and overestimate the performance of W and optimal policies, that inflates the expected benefit of the W and optimal policies. The naive bound (3) is unnecessarily wide, and cannot reliably infer the superiority of W and optimal policy even when $\Gamma = 2.0$. On the other hand, our method certifies the benefits of immediately administering antibiotics. For a larger level of confounding $\Gamma^{\star} = 5.0$, we explore the design sensitivity of our method and the naïve lower bound approach. Figure 1 (b,c) shows that for our method, the lower bound on

the performance of the W policy meets the upper bound on that of the WO policy at $\Gamma = 5.6$; our approach can reliably estimate that the W policy is better than the WO policy up to assuming an amount of confounding determined by $\Gamma = 5.6$. In contrast, the naïve bound has a a design sensitivity of $\Gamma = 1.75$, with bounds failing to be informative far below the true level of data confounding.

**Communication interventions for minimally verbal children with autism** Minimally verbal children represent 25-30% of children with autism, and often have poor prognosis on social functioning [49, 1]. Using a simulator for autistic children developed by Lu et al. [31], which models the data from a (real) sequential randomized trial (SMART) [23], we compare different approaches for improving the number of speech utterances. The potential interventions are a therapist's behavioral language intervention (BLI) or device for a augmented/alternative communication (AAC). Actions occur in weeks 0 and 12 ($T = 2$), and speech utterances are measured at weeks 0, 12, 24 and 36. The average number of speech utterances for a given patient as the reward/outcome. We consider a scenario where participants were randomly assigned to the two treatment options initially, and at time step 2, the clinician decides whether to add AAC devices for children who started with BLI. The supply of AAC devices is limited, so there may be a slight confounder of assignment by clinicians of AAC to children a clinician estimates are more likely to benefit, based on unrecorded patient interactions.

We compare an adaptive policy (BLI + AAC) that starts with BLI, and augments BLI with AAC at week 12 if the patient is a slow responder, against a non-adaptive policy (AAC) that uses AAC throughout; in the simulation, the BLI+AAC policy is worse than the AAC policy. OPE estimates for the AAC policy are unbiased since observations for this outcome are unconfounded. Figure 1 (d) shows standard OPE methods overestimate the outcome of the adaptive policy even at a low level of confounding $\Gamma^\star = 2.0$, incorrectly suggesting the BLI+AAC policy outperforms the AAC policy. Our lower bound on the adaptive policy computed using $\Gamma = 2.0$ suggests that the observed advantage of the adaptive policy may be solely due to confounding, even for small values of $\Gamma$.

# 6    Conclusion

For sequential problems, we analyzed the sensitivity of OPE methods to unobserved confounding in sequential decision making problems. We demonstrated how our approach can certify robustness of OPE in some settings, or raise concerns about its validity based on sensitivity to unobserved confounding. Our loss minimization method allows computing worst-case bounds over our bounded unobserved confounding model, while adjusting for observed features via importance sampling.

As a consequence, our estimators face the same challenges that standard importance-sampling-based OPE methods face: high variance when there is little overlap between the evaluation and behavior policy. In our experiments, importance sampling was effective since we ensured that there was sufficient overlap and focused on shorter horizons. In other settings, lack of overlap poses fundamental difficulties in off-policy evaluation, beyond issues with confounding, as others have also noted [7]. Such challenges become pronounced as the horizon $T$ or the importance sampling weights become large. While stationary importance sampling (SIS) can reduce variance, rewards under stationary distributions (should they exist) are not appropriate for the problems studied in this paper; SIS [9, 28, 56] nevertheless still suffers high variance when there is a lack of overlap. Fujimoto et al. [5], Kumar et al. [26] suggest some promising algorithmic approaches for only considering policies with sufficient overlap: while more work is needed, policies generated by these approaches would be more amenable to OPE, and should improve the statistical properties of our method.

It is natural to consider extending our single-decision confounding model to settings where a handful of decisions (say 2-5) are affected by unobserved confounding. Worst-case bounds on $\mathbb{E}[Y(\bar{A}_{1:T})]$ under such extensions require solving optimization problems involving products of likelihood ratios defined over different confounded time periods. Since these problems are nonconvex, they will require new approaches beyond our developments, which heavily depends on convex duality.

**Reproducibility** Our code is publicly available at `https://github.com/StanfordAI4HI/off_policy_confounding.git`

## Broader Impact

Understanding the impact of unobserved confounding in off-policy policy evaluation is crucial to reliable deployment of sequential policies. Our work is the one of the first—hopefully many others to follow—to develops tools that protect against deploying unreliable policies. For example, in healthcare systems, our methods can prevent premature deployment of unreliable automated policies based on spurious correlations, and guarantee that observed gains are robust to potential bias caused by unobserved confounding.

The goal of our method is to certify the reliability of off-policy policy evaluation. However, we require making assumptions about the data generating process to do so. In particular, we assume a bounded effect of confounding on the behavior policy's actions. Therefore, our certificates are only valid if this assumption holds. Users might be overconfident—trusting the bounds produced by this method could lead to problems if there is more confounding than assumed. To mitigate the risk of such usage, researchers should be aware of our modeling assumptions, have thoughtful conversations with application area experts about the plausible levels of confounding and think rigorously about all off-policy policy evaluation estimates and bounds. In high risk applications, results of our method or any off-policy policy evaluation should be confirmed in limited online evaluation before being widely deployed.

Finally, we hope that our work can bridge the gap between many different communities—epidemiology, economics, operations research, RL and causal inference—working on similar topics. Building reliable sequential policies for high stake scenarios like health care and education is an ambitious goal that will require an interdisciplinary effort.

## Acknowledgments and Disclosure of Funding

The research reported here was supported by NSF CAREER award and ONR Young Investigator Program.

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
