[Supplementary Material 1 · Supp.pdf]

# A Related Work

The dynamic treatment regime literature [41, 36] addressed many early questions around using observational data for sequential decision making, and developed a rich set of methods adapted for epidemiological questions. The reinforcement learning (RL) community is increasingly interested in developing theory and methods for the related problem of batch RL across a broad set of applications, because of new models and data availability (see e.g. [55, 29, 27, 54, 25, 11, 8]).

The majority of OPE methods for batch reinforcement learning rely on sequential ignorability (though often unstated). There is an extensive body of work for off-policy policy evaluation and optimization under this assumption, including doubly robust methods [16, 53] and recent work that provides semiparametric efficiency bounds [19]; often the behavior policy is assumed to be known. Notably, Liu et al. [29] highlights how estimation error in the behavioral policy can bias value estimates, and Nie et al. [37], Hanna et al. [10] provides OPE estimators based on an estimator of the behavior policy. When sequential ignorability doesn't hold, the expected cumulative rewards under an evaluation policy cannot be identified from observable data. All of the above estimators are biased in the presence of unobserved confounding, since neither the outcome model nor the importance sampling weights can correct for the effect of the unobserved confounder.

The do-calculus and its sequential backdoor criterion on the associated directed acyclic graph [39] gives identification results for OPE. Like sequential ignorability, this precludes the existence of unobserved confounding. Hence, methods that assume the sequential backdoor criterion will be biased in their presence.

We study the effects of unobserved confounding on OPE in sequential decision making problems, deriving bounds on the performance of the evaluation policy when sequential ignorability is relaxed. For problems where only one decision is made, a variety of methods developed in the econometrics, statistics, and epidemiology literature estimate bounds on treatment effects and expected rewards. Manski [33] developed bounds that only assume bounded rewards, though they are too conservative to identify whether one action is superior to another. Then, Manski [33] and other works posit models that bound the effect of unobserved confounding on the outcome [43, 3], or—like ours—on the actions taken by the behavior policy [4, 47, 15]. Recent work studied approaches that can apply to heterogeneous treatment effects [57, 22], policy evaluation [18], and policy optimization [20].

Tennenholtz et al. [52] studied OPE for partially observable Markov decision processes (POMDPs) and developed an identification strategy based on the independence structure of POMDP, similar to single decision work of [35]/not assume the existence of such variables or independence structures and seek to develop a lower bound on OPE. This also distinguishes our work from prior work which focuses on an algorithmic, scalable approach for when a single, time-invariant confounder is present, and which does not seek to present bounds on the OPE [30].

In sequential settings, Zhang and Bareinboim [58] derived partial identification bounds on policy performance with limited restrictions on the influence of the unobserved confounder on observed decisions, much like the single decision work of Manski [33], which they use to guide online RL algorithms. Unfortunately, these bounds can be too conservative to guide selection of policies. Robins et al. [43], Robins [42], Brumback et al. [3] instead posit a model for how confounding in each time step affects the outcome of interest and derive bounds under this model. Their work is motivated by potential confounding in the effects of dynamic treatment regimes for HIV therapy on CD4 counts in HIV-positive men. Our work is complementary to these in that we instead assume limited influence of the unobserved confounder on the behavior policy's actions.

Yadlowsky et al. [57] takes a similar approach as ours to bound the effect of confounding on treatment effects when there is only one action taken. Our approach allows for comparing sequences of actions derived according to an evaluation policy, by adjusting for the way actions in all time steps depend on the current states and history, and effect future states and rewards. One notable challenge that only occurs in sequential problems is adjusting for actions that occur after the confounded decision at time $t^\star$; these actions depend on the confounded decision through the history generated. A natural approach is to individually bound the potential outcomes $\mathbb{E}[Y(\bar{A}_{1:t^\star-1}, a_{t^\star:T})]$ for all $a_{t^\star:T}$, where each bound is given by a loss minimization problem. Under this approach—which is analogous to that of Yadlowsky et al. [57]—computing a lower bound to $\mathbb{E}[Y(\bar{A}_{1:T})]$ requires $\prod_{t=t^\star}^{T} |\mathcal{A}_t|$ loss minimization problems, making it statistically and computationally intractable when $t^\star$ is small (e.g. $t^\star = 1$ in our sepsis example). Instead, we consider averaged outcomes $\mathbb{E}[Y(\bar{A}_{1:t^\star-1}, a_{t^\star}, \bar{A}_{t^\star+1:T})]$

in Theorem 2, which allows us to obtain a lower bound on $\mathbb{E}[Y(\bar{A}_{1:T})]$ by solving $|\mathcal{A}_{t^\star}|$ loss minimization problems.

# B  Proof of basic lemmas

Before we give the proof of our main results, we give a set of essentially standard lemmas that we build on in the rest of the paper. In the following, we use a notational shorthand for (nested) expectations under observable distributions: for all $1 \le t \le T$ and $1 \le t_1 \le t_2 \le T$,

$$\mathbb{E}_{a_t}^t[X] := \mathbb{E}[X \mid H_t, A_t = a_t] \quad \text{and} \tag{5a}$$

$$\mathbb{E}_{a_{t_1:t_2}}^{t_1:t_2}[X] := \mathbb{E}_{a_{t_1}}^{t_1}[\mathbb{E}_{a_{t_1+1}}^{t_1+1}[\cdots \mathbb{E}_{a_{t_2}}^{t_2}[X]\cdots]]. \tag{5b}$$

Similarly, we write for all $1 \le t_0 \le t_1 \le t_2 \le T$

$$\mathbb{E}_{a_{t_1:t_2}}^{t_2}[X] := \mathbb{E}[X \mid H_t(A_{1:t_1-1}, a_{t_1:t_2-1}), A_{t_2} = a_{t_2}] \quad \text{and} \tag{6a}$$

$$\mathbb{E}_{a_{t_0:t_2}}^{t_1:t_2}[X] := \mathbb{E}_{a_{t_0:t_1}}^{t_1}[\mathbb{E}_{a_{t_0:t_1+1}}^{t_1+1}[\cdots \mathbb{E}_{a_{t_0:t_2}}^{t_2}[X]\cdots]]. \tag{6b}$$

The cumulative rewards $\mathbb{E}[Y(\bar{A}_{1:T})]$ under the candidate policy has an alternate representation, which we draw on heavily in the rest of the proofs. See Section B.1 for a derivation.

**Lemma 4.** *If sequential ignorability (Assumption B) holds for the evaluation policy $\bar{\pi}$, we have the identity*

$$\mathbb{E}\left[Y(\bar{A}_{1:T})\right] = \sum_{a_{1:T}} \mathbb{E}\left[Y(a_{1:T}) \prod_{t=1}^T \bar{\pi}_t(a_t \mid \bar{H}_t(a_{1:t-1}))\right].$$

To ease notation, denote each integrand in the above sum by

$$Y(a_{1:T}; \bar{\pi}) := Y(a_{1:T}) \prod_{t=1}^T \bar{\pi}_t(a_t \mid \bar{H}_t(a_{1:t-1})). \tag{7}$$

We will also use the following two identities heavily. Recall that we denote by $W := \{W(a_{1:T})\}_{a_{1:T}}$, the tuple of all potential outcomes, which takes values in $\mathcal{W}$. See Section B.2 for a proof of the following result.

**Lemma 5.** *Let sequential ignorability (Assumption B) hold for the behavioral policy $\pi$ in the time steps $t_1 : t_2$, where $1 \le t_1 < t_2 \le T$. Then, for any measurable $f : \mathcal{W} \to \mathbb{R}$*

$$\mathbb{E}[f(W) \mid H_{t_1}(a_{1:t_1-1})] = \mathbb{E}\left[\mathbb{E}_{a_{1:t_2}}^{t_1:t_2}[f(W)] \mid H_{t_1}(a_{1:t_1-1})\right]$$

*for any $a_{1:t_2} \in \mathcal{A}_1 \times \cdots \times \mathcal{A}_{t_2}$.*

The following identity—whose proof we give in Section B.3—is a simple consequence of the definition of conditional expectations, and the tower law.

**Lemma 6.** *For any measurable function $f : \mathcal{W} \to \mathbb{R}$, and $1 \le t_1 \le t_2 \le T$,*

$$\mathbb{E}_{a_{1:t_2}}^{t_1:t_2} f(W) = \mathbb{E}\left[f(W) \prod_{t=t_1}^{t_2} \frac{\mathbb{1}\{A_T = a_t\}}{\pi_t(a_t \mid H_t(a_{1:t-1}))} \mid H_{t_1}(a_{1:t_1-1})\right]$$

## B.1  Proof of Lemma 4

Similar to the notational shorthand (5), define

$$\overline{\mathbb{E}}_{a_{1:t}}^t[X] := \mathbb{E}[X \mid \bar{H}_t(a_{1:t-1}), \bar{A}_t = a_t] \quad \text{and} \quad \overline{\mathbb{E}}_{a_{1:T}}^{t:T}[X] = \overline{\mathbb{E}}_{a_{1:t}}^t[\overline{\mathbb{E}}_{a_{1:t+1}}^{t+1}[\cdots \overline{\mathbb{E}}_{a_{1:T}}^T[X]\cdots]].$$

Begin by noting that by definition of conditional expectation

$$\mathbb{E}[Y(\bar{A}_{1:T}) \mid \bar{H}_1] = \sum_{a_1 \in \mathcal{A}_1} \bar{\pi}(a_1 \mid \bar{H}_1) \mathbb{E}[Y(a_1, \bar{A}_{2:T}) \mid \bar{H}_1, \bar{A}_1 = a_1]$$

$$= \sum_{a_1 \in \mathcal{A}_1} \bar{\pi}(a_1 \mid \bar{H}_1) \overline{\mathbb{E}}_{a_1}^1[Y(a_1, \bar{A}_{2:T})],$$

and similarly, conditioning on $\bar{H}_2(a_1) = (S_1, \bar{A}_1 = a_1, S_2(a_1))$ yields

$$\mathbb{E}[Y(a_1, \bar{A}_{2:T}) \mid \bar{H}_2(a_1)] = \sum_{a_2 \in \mathcal{A}_2} \bar{\pi}_2(a_2 \mid \bar{H}_2(a_1))\mathbb{E}[Y(a_{1:2}, \bar{A}_{3:T}) \mid \bar{H}_2(a_1), \bar{A}_2 = a_2]$$

$$= \sum_{a_2 \in \mathcal{A}_2} \bar{\pi}_2(a_2 \mid \bar{H}_2(a_1))\overline{\mathbb{E}}^2_{a_{1:2}}[Y(a_{1:2}, \bar{A}_{3:T})].$$

From the tower law, the above two equalities yield

$$\mathbb{E}[Y(\bar{A}_{1:T})] = \mathbb{E}\left[\sum_{a_1 \in \mathcal{A}_1} \bar{\pi}_1(a_1 \mid \bar{H}_1)\mathbb{E}\left[\sum_{a_2 \in \mathcal{A}} \bar{\pi}_2(a_2 \mid \bar{H}_2(a_1)) \cdot \mathbb{E}[Y(a_{1:2}, \bar{A}_{3:T}) \mid \bar{H}_2(a_1), \bar{A}_2 = a_2]\,\Big|\,\bar{H}_1, \bar{A}_1 = a_1\right]\right]$$

$$= \mathbb{E}\left[\sum_{a_1 \in \mathcal{A}_1} \bar{\pi}_1(a_1 \mid \bar{H}_1)\overline{\mathbb{E}}^1_{a_1}\left[\sum_{a_2 \in \mathcal{A}} \bar{\pi}_2(a_2 \mid \bar{H}_2(a_1)) \cdot \overline{\mathbb{E}}^2_{a_{1:2}}[Y(a_{1:2}, \bar{A}_{3:T})]\right]\right].$$

Proceeding iteratively as before and expanding each $\mathbb{E}[Y(a_{1:t-1}, \bar{A}_{t:T}) \mid \bar{H}_t(a_{1:t-1})]$, we arrive at

$$\mathbb{E}[Y(\bar{A}_{1:T})] =$$
$$\mathbb{E}\left[\sum_{a_1 \in \mathcal{A}_1} \bar{\pi}_1(a_1 \mid \bar{H}_1)\overline{\mathbb{E}}^1_{a_1}\left[\overline{\mathbb{E}}^2_{a_{1:2}}\left[\sum_{a_2 \in \mathcal{A}_2} \bar{\pi}_2(a_2 \mid \bar{H}_2(a_1))\overline{\mathbb{E}}^3_{a_{1:3}} \times \left[\cdots \sum_{a_T \in \mathcal{A}_T} \bar{\pi}_T(a_T \mid \bar{H}_T(a_{1:T-1}))\overline{\mathbb{E}}^T_{a_{1:T}}[Y(a_{1:T})]\right]\right]\right]\right].$$

Now, we proceed backwards from the inner most expectation to take the outer sum inside the expectation. By Assumption B, we have

$$\sum_{a_T \in \mathcal{A}_T} \bar{\pi}_T(a_T \mid \bar{H}_T(a_{1:T-1}))\overline{\mathbb{E}}^T_{a_{1:T}}[Y(a_{1:T})] = \sum_{a_T \in \mathcal{A}_T} \bar{\pi}_T(a_T \mid \bar{H}_T(a_{1:T-1})) \cdot \mathbb{E}\left[Y(a_{1:T})\,\Big|\,\bar{H}_T(a_{1:T-1})\right]$$

$$= \mathbb{E}\left[\sum_{a_T \in \mathcal{A}_T} \bar{\pi}_T(a_T \mid \bar{H}_T(a_{1:T-1})) \cdot Y(a_{1:T})\,\Big|\,\bar{H}_T(a_{1:T-1})\right].$$

Noting that $\mathbb{E}[\cdot \mid \bar{H}_T(a_{1:T-1})] = \mathbb{E}[\cdot \mid \bar{H}_{T-1}(a_{1:T-2}), S_T(a_{1:T-1}), \bar{A}_{T-1}=a_{T-1}]$, the tower law and preceding display yield

$$\overline{\mathbb{E}}^{T-1}_{a_{1:T-1}}\left[\sum_{a_T \in \mathcal{A}_T} \bar{\pi}_T(a_T \mid \bar{H}_T(a_{1:T-1})) \cdot \overline{\mathbb{E}}^T_{a_{1:T}}[Y(a_{1:T})]\right] = \overline{\mathbb{E}}^{T-1}_{a_{1:T-1}}\left[\sum_{a_T \in \mathcal{A}_T} \bar{\pi}_T(a_T \mid \bar{H}_T(a_{1:T-1})) \cdot Y(a_{1:T})\right].$$

We repeat an identical process for the sum over $a_{T-1}$. Similarly as above, applying Assumption B gives

$$\sum_{a_{T-1} \in \mathcal{A}_{T-1}} \bar{\pi}_{T-1}(a_{T-1} \mid \bar{H}_{T-1}(a_{1:T-2})) \cdot \overline{\mathbb{E}}^{T-1}_{a_{1:T-1}}\left[\sum_{a_T \in \mathcal{A}_T} \bar{\pi}_T(a_T \mid \bar{H}_T(a_{1:T-1})) \cdot Y(a_{1:T})\right]$$

$$= \mathbb{E}\left[\sum_{a_{T-1} \in \mathcal{A}_{T-1}} \bar{\pi}_{T-1}(a_{T-1} \mid \bar{H}_{T-1}(a_{1:T-2})) \sum_{a_T \in \mathcal{A}_T} \bar{\pi}_T(a_T \mid \bar{H}_T(a_{1:T-1})) \cdot Y(a_{1:T})\,\Big|\,\bar{H}_{T-1}(a_{1:T-2})\right].$$

By the tower law, we again get

$$\overline{\mathbb{E}}^{T-2}_{a_{1:T-2}}\left[\sum_{a_{T-1} \in \mathcal{A}_{T-1}} \bar{\pi}_{T-1}(a_{T-1} \mid \bar{H}_{T-1}(a_{1:T-2}))\overline{\mathbb{E}}^{T-1}_{a_{1:T-1}}\left[\sum_{a_T \in \mathcal{A}_T} \bar{\pi}_T(a_T \mid \bar{H}_T(a_{1:T-1})) \cdot Y(a_{1:T})\right]\right]$$

$$= \overline{\mathbb{E}}^{T-2}_{a_{1:T-2}}\left[\sum_{a_{T-1} \in \mathcal{A}_{T-1}} \bar{\pi}_{T-1}(a_{T-1} \mid \bar{H}_{T-1}(a_{1:T-2})) \cdot \sum_{a_T \in \mathcal{A}_T} \bar{\pi}_T(a_T \mid \bar{H}_T(a_{1:T-1})) \cdot Y(a_{1:T})\right].$$

Iterating the above process over the indices $t = T - 2, \ldots, 1$, we arrive at the desired formula.

## B.2 Proof of Lemma 5

From the tower law and sequential ignorability of $\pi$,

$$\mathbb{E}[f(W) \mid H_{t_1}(a_{1:t_1-1})] = \mathbb{E}[f(W) \mid H_{t_1}(a_{1:t_1-1}), A_{t_1} = a_{t_1}]$$
$$= \mathbb{E}[\mathbb{E}[f(W) \mid H_{t_1+1}(a_{1:t_1})] \mid H_{t_1}(a_{1:t_1-1}), A_{t_1} = a_{t_1}]$$

Applying the tower law to the inner expectation, and applying sequential ignorability again, we get

$$\mathbb{E}[f(W) \mid H_{t_1+1}(a_{1:t_1})] = \mathbb{E}\left[\mathbb{E}[f(W) \mid H_{t_1+2}(a_{1:t_1+1})] \mid H_{t_1+1}(a_{1:t_1}), A_{t_1+1} = a_{t_1+1}\right]$$

Plugging this back into the original display, we have

$$\mathbb{E}[f(W) \mid H_{t_1}(a_{1:t_1-1})] = \mathbb{E}_{a_{1:t_1+1}}^{t_1:t_1+1}\left[\mathbb{E}[f(W) \mid H_{t_1+2}(a_{1:t_1+1})]\right]$$

Repeating this argument over $t = t_1 + 2, \ldots, t_2$, we conclude the result.

## B.3 Proof of Lemma 6

From the definition of conditional expectations, we have

$$\mathbb{E}[f(W) \mid H_t(a_{1:t-1}), A_t = a_t] = \mathbb{E}\left[f(W)\frac{\mathbb{1}\{A_t = a_t\}}{\pi_t(a_t \mid H_t(a_{1:t-1}))} \mid H_t(a_{1:t-1})\right].$$

The result follows by applying this equality at $t = t_2$, applying the tower law, and iterating the same argument over $t = t_2 - 1, \ldots, t_1$.

# C  Proof of key identities

## C.1 Proof of Lemma 1

Recalling the notation (7), sequential ignorability of $\bar{\pi}$ and Lemma 4 gives the following representation

$$\mathbb{E}\left[Y(\bar{A}_{1:T})\right] = \sum_{a_{1:T}} \mathbb{E}\left[Y(a_{1:T}; \bar{\pi})\right].$$

We deal with each term $\mathbb{E}[Y(a_{1:T}; \bar{\pi})]$ in the summation separately, for each fixed sequence of actions $a_{1:T}$. From sequential ignorability of $\pi$ and Lemma 5,

$$\mathbb{E}[Y(a_{1:T}; \bar{\pi})] = \mathbb{E}[\mathbb{E}_{a_1}^1[\cdots \mathbb{E}_{a_{1:T}}^T[Y(a_{1:T}; \bar{\pi})]\cdots]] = \mathbb{E}[\mathbb{E}_{a_{1:T}}^{1:T}[Y(a_{1:T}; \bar{\pi})]].$$

Applying Lemma 6, we get

$$\mathbb{E}[Y(a_{1:T}; \bar{\pi})] = \mathbb{E}\left[Y(a_{1:T}; \bar{\pi})\prod_{t=1}^{T}\frac{\mathbb{1}\{A_t = a_t\}}{\pi_t(a_t \mid H_t(a_{1:t-1}))}\right].$$

Summing the preceeding display over $a_{1:T}$, we obtain the desired result.

## C.2 Proof of Proposition 1

From Lemma 4, we have

$$\mathbb{E}[Y(\bar{A}_{1:T})] = \mathbb{E}\left[\sum_{a_{1:T}} Y(a_{1:T})\prod_{t=1}^{T}\bar{\pi}_t(a_t \mid \bar{H}_t(a_{1:t-1}))\right].$$

Since sequential ignorability for $\pi$ holds at any $t < t^\star$, Lemma 5 implies that the preceeding display is equal to

$$\mathbb{E}\left[\sum_{a_{1:t^\star-1}} \mathbb{E}_{a_{1:t^\star-1}}^{1:t^\star-1}\left[\sum_{a_{t^\star:T}} Y(a_{1:T})\prod_{t=1}^{T}\bar{\pi}_t(a_t \mid \bar{H}_t(a_{1:t-1}))\right]\right].$$

Applying Lemma 6 to the inner expectations, we get

$$\mathbb{E}[Y(\bar{A}_{1:T})] = \mathbb{E}\left[\sum_{a_{1:t^\star-1}}\prod_{t=1}^{t^\star-1}\frac{\mathbb{1}\{A_t = a_t\}}{\pi_t(a_t \mid H_t(a_{1:t-1}))}\sum_{a_{t^\star:T}}Y(a_{1:T})\prod_{t=1}^{T}\bar{\pi}_t(a_t \mid \bar{H}_t(a_{1:t-1}))\right]$$

$$= \mathbb{E}\left[\prod_{t=1}^{t^\star-1}\frac{\bar{\pi}_t(A_t \mid \bar{H}_t(A_{1:t-1}))}{\pi_t(A_t \mid H_t)}\sum_{a_{t^\star:T}}Y(A_{1:t^\star-1}, a_{t^\star:T})\prod_{t=t^\star}^{T}\bar{\pi}_t(a_t \mid \bar{H}_t(A_{1:t^\star-1}, a_{t^\star:t-1}))\right].$$

From the tower law, we arrive at

$$\mathbb{E}[Y(\bar{A}_{1:T})] = \mathbb{E}\left[\mathbb{E}\left[\prod_{t=1}^{t^\star-1}\frac{\bar{\pi}_t(A_t \mid \bar{H}_t(A_{1:t-1}))}{\pi_t(A_t \mid H_t)}\sum_{a_{t^\star:T}}Y(A_{1:t^\star-1}, a_{t^\star:T})\prod_{t=t^\star}^{T}\bar{\pi}_t(a_t \mid \bar{H}_t(A_{1:t^\star-1}, a_{t^\star:t-1}))\,\Bigg|\, H_{t^\star}\right]\right]$$

$$= \mathbb{E}\left[\prod_{t=1}^{t^\star-1}\frac{\bar{\pi}_t(A_t \mid \bar{H}_t(A_{1:t-1}))}{\pi_t(A_t \mid H_t)}\mathbb{E}\left[\sum_{a_{t^\star:T}}Y(A_{1:t^\star-1}, a_{t^\star:T})\prod_{t=t^\star}^{T}\bar{\pi}_t(a_t \mid \bar{H}_t(A_{1:t^\star-1}, a_{t^\star:t-1}))\,\Bigg|\, H_{t^\star}\right]\right].$$

$$(9)$$

Applying the tower law to the inner expectation in the final display, we can write

$$\mathbb{E}\left[\sum_{a_{t^\star:T}}Y(A_{1:t^\star-1}, a_{t^\star:T})\prod_{t=t^\star}^{T}\bar{\pi}_t(a_t \mid \bar{H}_t(A_{1:t^\star-1}, a_{t^\star:t-1}))\,\Bigg|\, H_{t^\star}\right]$$

$$= \mathbb{E}\left[\mathbb{E}\left[\sum_{a_{t^\star:T}}Y(A_{1:t^\star-1}, a_{t^\star:T})\prod_{t=t^\star}^{T}\bar{\pi}_t(a_t \mid \bar{H}_t(A_{1:t^\star-1}, a_{t^\star:t-1}))\,\Bigg|\, H_{t^\star}, A_{t^\star}\right]\,\Bigg|\, H_{t^\star}\right]$$

$$= \sum_{a_{t^\star}, a'_{t^\star}}\bar{\pi}_{t^\star}(a_{t^\star} \mid \bar{H}_{t^\star}(A_{1:t^\star-1}))\pi_{t^\star}(a'_{t^\star} \mid H_{t^\star})$$

$$\times \mathbb{E}\left[\sum_{a_{t^\star+1:T}}Y(A_{1:t^\star-1}, a_{t^\star:T})\prod_{t=t^\star+1}^{T}\bar{\pi}_t(a_t \mid \bar{H}_t(A_{1:t^\star-1}, a_{t^\star:t-1}))\,\Bigg|\, H_{t^\star}, A_{t^\star} = a'_{t^\star}\right]$$

$$= \sum_{a_{t^\star}, a'_{t^\star}}\bar{\pi}_{t^\star}(a_{t^\star} \mid \bar{H}_{t^\star}(A_{1:t^\star-1}))\pi_{t^\star}(a'_{t^\star} \mid H_{t^\star})$$

$$\times \mathbb{E}\left[\mathcal{L}(W; H_{t^\star}, a_{t^\star}, a'_{t^\star})\sum_{a_{t^\star+1:T}}Y(A_{1:t^\star-1}, a_{t^\star:T})\prod_{t=t^\star+1}^{T}\bar{\pi}_t(a_t \mid \bar{H}_t(A_{1:t^\star-1}, a_{t^\star:t-1}))\,\Bigg|\, H_{t^\star}, A_{t^\star} = a_{t^\star}\right]$$

where in the last equality, we used the definition

$$\mathcal{L}(\cdot; H_{t^\star}, a_{t^\star}, a'_{t^\star}) := \frac{dP_W(\cdot \mid H_{t^\star}, A_{t^\star} = a'_{t^\star})}{dP_W(\cdot \mid H_{t^\star}, A_{t^\star} = a_{t^\star})}.$$

Again, by the tower law,

$$\mathbb{E}\left[\mathcal{L}(W; H_{t^\star}, a_{t^\star}, a'_{t^\star})\sum_{a_{t^\star+1:T}}Y(A_{1:t^\star-1}, a_{t^\star:T})\prod_{t=t^\star+1}^{T}\bar{\pi}_t(a_t \mid \bar{H}_t(A_{1:t^\star-1}, a_{t^\star:t-1}))\,\Bigg|\, H_{t^\star}, A_{t^\star} = a_{t^\star}\right]$$

$$= \mathbb{E}\left[\mathbb{E}\left[\mathcal{L}(W; H_{t^\star}, a_{t^\star}, a'_{t^\star})\sum_{a_{t^\star+1:T}}Y(A_{1:t^\star-1}, a_{t^\star:T})\right.\right.$$

$$\left.\left. \times \prod_{t=t^\star+1}^{T}\bar{\pi}_t(a_t \mid \bar{H}_t(A_{1:t^\star-1}, a_{t^\star:t-1}))\Big|H_{t^\star+1}(A_{1:t^\star-1}, a_{t^\star})\right]\,\Bigg|\, H_{t^\star}, A_{t^\star} = a_{t^\star}\right]$$

From sequential ignorability of $\pi$ for $t > t^\star$ and Lemma 5, the preceeding display is equal to

$$\mathbb{E}\left[\sum_{a_{t^\star+1:T}}\mathbb{E}_{a_{t^\star:T}}^{t^\star+1:T}\mathcal{L}(W; H_{t^\star}, a_{t^\star}, a'_{t^\star})Y(A_{1:t^\star-1}, a_{t^\star:T})\prod_{t=t^\star+1}^{T}\bar{\pi}_t(a_t \mid \bar{H}_t(A_{1:t^\star-1}, a_{t^\star:t-1}))\,\Bigg|\, H_{t^\star}, A_{t^\star} = a_{t^\star}\right].$$

From Lemma 6, we can rewrite the above expression as

$$\mathbb{E}\left[\mathcal{L}(W; H_{t^\star}, a_{t^\star}, a'_{t^\star}) Y_{t^\star}(a_{t^\star}) \prod_{t=t^\star+1}^{T} \frac{\bar{\pi}_t(A_t \mid \bar{H}_t(A_{1:t^\star-1}, a_{t^\star}, A_{t^\star+1:t}))}{\pi_t(A_t \mid H_t(A_{1:t^\star-1}, a_{t^\star}, A_{t^\star+1:t}))} \;\middle|\; H_{t^\star}, A_{t^\star} = a_{t^\star}\right].$$

Plugging these expressions back into the equality (9), we obtain the result.

## D   Proof of bounds under unobserved confounding

### D.1   Naive bound

We show the below more general result.

**Lemma 7.** *Let Assumptions B, C, D hold. Then, we have*

$$\mathbb{E}[Y(\bar{A}_{1:T})] \geq \mathbb{E}\left[Y(A_{1:T}) \prod_{t=1}^{T} \frac{\bar{\pi}_t(A_t \mid \bar{H}_t(A_{1:t-1}))}{\pi_t(A_t \mid H_t)(\Gamma_t^{-1}\mathbb{1}\{Y(A_{1:T}) < 0\} + \Gamma_t\mathbb{1}\{Y(A_{1:T}) > 0\})}\right].$$

**Proof of Lemma**     From an identical argument as the proof of Lemma 1, Assumption C yields

$$\mathbb{E}[Y(\bar{A}_{1:T})] = \mathbb{E}\left[Y(A_{1:T}) \prod_{t=1}^{T} \frac{\bar{\pi}_t(A_t \mid \bar{H}_t(A_{1:t-1}), U_t)}{\pi_t(A_t \mid H_t, U_t)}\right].$$

From Assumption B, the preceeding display is equal to

$$\mathbb{E}[Y(\bar{A}_{1:T})] = \mathbb{E}\left[Y(A_{1:T}) \prod_{t=1}^{T} \frac{\bar{\pi}_t(A_t \mid \bar{H}_t(A_{1:t-1}))}{\pi_t(A_t \mid H_t, U_t)}\right]. \tag{10}$$

Now, we bound $\pi_t(A_t \mid H_t, U_t)$ by $\pi_t(A_t \mid H_t)$. Assumption D implies

$$\pi_t(a_t \mid H_t, U_t = u_t)\pi_t(a'_t \mid H_t, U_t = u'_t) \leq \Gamma_t \pi_t(a'_t \mid H_t, U_t = u_t)\pi_t(a_t \mid H_t, U_t = u'_t).$$

Multiplying by $p_{U_t}(u'_t \mid H_t)$ on both sides and integrating over $u'_t$, we get

$$\pi_t(a_t \mid H_t, U_t = u_t)\pi_t(a'_t \mid H_t) \leq \Gamma_t \pi_t(a'_t \mid H_t, U_t = u_t)\pi_t(a_t \mid H_t).$$

Summing over $a'_t$ on both sides, we conclude that

$$\pi_t(a_t \mid H_t, U_t = u_t) \leq \Gamma_t \pi_t(a_t \mid H_t).$$

almost surely, for any $t, a_t, H_t, u_t$. Using this relation to lower bound expression (10), we obtain the result.     □

### D.2   Proof of Theorem 2

By rewriting the original infimization problem over $L(W; H_{t^\star})$ to $L(W, A_{t^\star+1:T}; H_{t^\star})$, we have
$\eta^\star(H_{t^\star}; a_{t^\star}) =$

$$\inf_{L \geq 0}\left\{\mathbb{E}\left[L(W, A_{t^\star+1:T}; H_{t^\star})Y(A_{1:T}) \prod_{t=t^\star+1}^{T} \rho_t \;\middle|\; H_{t^\star}, A_{t^\star} = a_{t^\star}\right] : \mathbb{E}[L(W, A_{t^\star+1:T}; H_{t^\star}) \mid H_{t^\star}, A_{t^\star} = a_{t^\star}] = 1, \text{ and}\right.$$

$$\left. L(w, a_{t^\star+1:T}; H_{t^\star}) = L(w, a'_{t^\star+1:T}; H_{t^\star}), \quad L(w, a_{t^\star+1:T}; H_{t^\star}) \leq \Gamma L(w', a'_{t^\star+1:T}; H_{t^\star}) \text{ a.s. all } w, a_{t^\star+1:T}, w', a'_{t^\star+1:T}\right\}.$$

Relaxing the equality constraint $L(w, a_{t^\star+1:T}; H_{t^\star}) = L(w, a'_{t^\star+1:T}; H_{t^\star})$, we arrive at
$\eta^\star(H_{t^\star}; a_{t^\star}) \geq$

$$\inf_{L \geq 0}\left\{\mathbb{E}\left[L(W, A_{t^\star+1:T}; H_{t^\star})Y(A_{1:T}) \prod_{t=t^\star+1}^{T} \rho_t \;\middle|\; H_{t^\star}, A_{t^\star} = a_{t^\star}\right] : \right.$$

$$\left. \mathbb{E}[L(W, A_{t^\star+1:T}; H_{t^\star}) \mid H_{t^\star}, A_{t^\star} = a_{t^\star}] = 1, \text{ and } L(w, a_{t^\star+1:T}; H_{t^\star}) \leq \Gamma L(w', a'_{t^\star+1:T}; H_{t^\star}) \text{ a.s. all } w, a_{t^\star+1:T}, w', a'_{t^\star+1:T}\right\}.$$

The preceeding optimization problem is convex, and Slater's condition holds for $L \equiv 1$. By strong duality [32, Thm. 8.6.1 and Problem 8.7], we obtain the dual formulation

$$\sup_{\mu} \inf_{L \geq 0} \left\{ \mathbb{E}\left[ L(W, A_{t^\star+1:T}; H_{t^\star}) \left( Y(A_{1:T}) \prod_{t=t^\star+1}^{T} \rho_t - \mu \right) \Bigg| H_{t^\star}, A_{t^\star} = a_{t^\star} \right] + \mu : \right.$$
$$\left. L(w, a_{t^\star+1:T}; H_{t^\star}) \leq \Gamma L(w', a'_{t^\star+1:T}; H_{t^\star}) \text{ a.s. all } w, a_{t^\star+1:T}, w', a'_{t^\star+1:T} \right\}.$$

By inspection, the solution to the inner infimum takes the form

$$L(w, a_{t^\star+1:T}; H_{t^\star}) = c \left( \Gamma \mathbb{1}\left\{ Y(A_{1:T}) \prod_{t=t^\star+1}^{T} \rho_t - \mu < 0 \right\} + \mathbb{1}\left\{ Y(A_{1:T}) \prod_{t=t^\star+1}^{T} \rho_t - \mu \geq 0 \right\} \right)$$

for some constant $c > 0$. Let $\ell'_\Gamma(z) := (z)_+ - \Gamma(z)_-$, the derivative of the weighted squared loss $\ell_\Gamma(z) = \frac{1}{2}(\Gamma(z)_-^2 + (z)_+^2)$. Plugging the preceeding display into the dual formulation, we get

$$\sup_{\mu} \inf_{c \geq 0} \left\{ c\mathbb{E}\left[ \ell'_\Gamma\left( Y(A_{1:T}) \prod_{t=t^\star+1}^{T} \rho_t - \mu \right) \Bigg| H_{t^\star}, A_{t^\star} = a_{t^\star} \right] + \mu \right\}$$
$$= \sup_{\mu} \left\{ \mu : \mathbb{E}\left[ \ell'_\Gamma\left( Y(A_{1:T}) \prod_{t=t^\star+1}^{T} \rho_t - \mu \right) \Bigg| H_{t^\star}, A_{t^\star} = a_{t^\star} \right] \geq 0 \right\}.$$

Since the function $\mu \mapsto \mathbb{E}\left[ \ell'_\Gamma\left( Y(A_{1:T}) \prod_{t=t^\star+1}^{T} \rho_t - \mu \right) \Big| H_{t^\star}, A_{t^\star} = a_{t^\star} \right]$ is strictly decreasing, the optimal solution (and its value) in the preceeding display is given by the unique zero of this function.

We now show that the solution to our loss minimization problem

$$\kappa(H_{t^\star}; a_{t^\star}) = \operatorname*{argmin}_{f(H_{t^\star})} \left\{ \mathbb{E}\left[ \frac{\mathbb{1}\{A_{t^\star} = a_{t^\star}\}}{\pi_{t^\star}(a_{t^\star} \mid H_{t^\star})} \times \ell_\Gamma\left( Y(A_{1:T}) \prod_{t=t^\star+1}^{T} \rho_t - f(H_{t^\star}) \right) \right] \right.$$
$$\left. = \mathbb{E}\left[ \mathbb{E}\left[ \ell_\Gamma\left( Y(A_{1:T}) \prod_{t=t^\star+1}^{T} \rho_t - f(H_{t^\star}) \right) \Bigg| H_{t^\star}, A_{t^\star} = a_{t^\star} \right] \right] \right\}$$

is in fact the unique zero of the function $\mu \mapsto \mathbb{E}\left[ \ell'_\Gamma\left( Y(A_{1:T}) \prod_{t=t^\star+1}^{T} \rho_t - \mu \right) \Big| H_{t^\star}, A_{t^\star} = a_{t^\star} \right]$. The (almost sure) uniqueness of the solution follows from strong convexity of $\ell_\Gamma$. Since the optimization is over all $H_{t^\star}$-measurable functions, the argmin is given by

$$\operatorname*{argmin}_{f(H_{t^\star})} \mathbb{E}\left[ \ell_\Gamma\left( Y(A_{1:T}) \prod_{t=t^\star+1}^{T} \rho_t - f(H_{t^\star}) \right) \Bigg| H_{t^\star}, A_{t^\star} = a_{t^\star} \right].$$

So long as $\mathbb{E}[Y(A_{1:T})^2 \prod_{t=t^\star+1}^{T} \rho_t^2 \mid A_{t^\star} = a_{t^\star}, H_{t^\star}] < \infty$ almost surely, first order optimality conditions of this loss minimization problem is equivalent to $\mathbb{E}\left[ \ell'_\Gamma\left( Y(A_{1:T}) \prod_{t=t^\star+1}^{T} \rho_t - f(H_{t^\star}) \right) \Big| H_{t^\star}, A_{t^\star} = a_{t^\star} \right] = 0$, which gives our result.

### D.3 Proof of Theorem 3

Our result is based on epi-convergence theory [24, 44], which shows (uniform) convergence of convex functions, and solutions to convex optimization problems.

**Definition 2.** *Let* $\{A_n\}$ *be a sequence of subsets of* $\mathbb{R}^d$. *The* limit supremum *(or* limit exterior *or* outer limit*) and* limit infimum *(*limit interior *or* inner limit*) of the sequence* $\{A_n\}$ *are*

$$\limsup_n A_n := \left\{ v \in \mathbb{R}^d \mid \liminf_{n \to \infty} \mathrm{dist}(v, A_n) = 0 \right\} \quad and$$

$$\liminf_n A_n := \left\{ v \in \mathbb{R}^d \mid \limsup_{n \to \infty} \mathrm{dist}(v, A_n) = 0 \right\}.$$

The epigraph of a function $h : \mathbb{R}^d \to \mathbb{R} \cup \{+\infty\}$ is $\mathrm{epi}\, h := \{(x, t) \in \mathbb{R}^d \times \mathbb{R} \mid h(x) \leq t\}$. We say $\lim_n A = A_\infty$ if $\limsup_n A_n = \liminf_n A_n = A_\infty \subset \mathbb{R}^d$. We define a notion of convergence for functions in terms of their epigraphs.

**Definition 3.** *A sequence of functions* $h_n$ *epi-converges to a function* $h$, *denoted* $h_n \overset{\mathrm{epi}}{\to} h$, *if*

$$\mathrm{epi}\, h = \liminf_{n \to \infty} \mathrm{epi}\, h_n = \limsup_{n \to \infty} \mathrm{epi}\, h_n. \tag{11}$$

If $h$ is proper ($\mathrm{dom}\, h \neq \varnothing$), epigraphical convergence (11) is characterizaed by pointwise convergence on a dense set.

**Lemma 8** (Theorem 7.17, Rockafellar and Wets [44]). *Let* $h_n : \mathbb{R}^d \to \overline{\mathbb{R}}, h : \mathbb{R}^d \to \overline{\mathbb{R}}$ *be closed, convex, and proper. Then* $h_n \overset{\mathrm{epi}}{\to} h$ *is equivalent to either of the following two conditions.*

*(i) There exists a dense set* $A \subset \mathbb{R}^d$ *such that* $h_n(v) \to h(v)$ *for all* $v \in A$.

*(ii) For all compact* $C \subset \mathrm{dom}\, h$ *not containing a boundary point of* $\mathrm{dom}\, h$,

$$\lim_{n \to \infty} \sup_{v \in C} |h_n(v) - h(v)| = 0.$$

The last characterization of epigraphical convergence is powerful as it gives convergence of solution sets.

**Lemma 9** (Theorem 7.31, Rockafellar and Wets [44]). *Let* $h_n : \mathbb{R}^d \to \overline{\mathbb{R}}, h : \mathbb{R}^d \to \overline{\mathbb{R}}$ *satisfy* $h_n \overset{\mathrm{epi}}{\to} h$ *and* $-\infty < \inf h < \infty$. *Let* $S_n(\varepsilon) = \{\theta \mid h_n(\theta) \leq \inf h_n + \varepsilon\}$ *and* $S(\varepsilon) = \{\theta \mid h(\theta) \leq \inf h + \varepsilon\}$. *Then* $\limsup_n S_n(\varepsilon) \subset S(\varepsilon)$ *for all* $\varepsilon \geq 0$, *and* $\limsup_n S_n(\varepsilon_n) \subset S(0)$ *whenever* $\varepsilon_n \downarrow 0$.

From Lemmas 8, 9, it suffices to show that the expected loss function and its empirical counterpart satisfies appropriate regularity conditions (proper and closed), and show that our empirical loss pointwise converges to the population loss almost surely. Recall that $\mathcal{D}_n$ is the split of data used to estimate $\widehat{\pi}$, and let $\mathcal{D}_\infty$ be the $\sigma$-algebra defined by $\mathcal{D}_n$ as $n \to \infty$. Our subsequent argument will be conditional on $\mathcal{D}_\infty$, and the event

$$\mathcal{E} := \left\{ \widehat{\pi}_t \to \pi_t \text{ pointwise}, \ \widehat{\rho}_t \leq 2C, \text{ and } \widehat{\pi}_{t^\star}(a_{t^\star} \mid H_{t^\star}) \in [(2C)^{-1}, 1] \right\}.$$

We assume w.l.o.g. (increasing $C$ if necessary) that $c \leq (2C)^{-1}$. Note that $\mathbb{P}(\mathcal{E}) = 1$ by assumption.

First, note that since $\theta \mapsto f_\theta$ is linear, $\theta \mapsto \ell_\Gamma(Y(A_{1:T}) \prod_{t=t^\star+1}^T \widehat{\rho}_t - f_\theta(H_{t^\star}))$ is convex. Both the empirical and population loss

$$\theta \mapsto \widehat{\mathbb{E}}_n \left[ \frac{\mathbb{1}\{A_{t^\star} = a_{t^\star}\}}{\widehat{\pi}_{t^\star}(a_{t^\star} \mid H_{t^\star})} \ell_\Gamma \left( Y(A_{1:T}) \prod_{t=t^\star+1}^T \widehat{\rho}_t - f_\theta(H_{t^\star}) \right) \right] =: \widehat{h}_n(\theta),$$

$$\theta \mapsto \mathbb{E} \left[ \frac{\mathbb{1}\{A_{t^\star} = a_{t^\star}\}}{\pi_{t^\star}(a_{t^\star} \mid H_{t^\star})} \ell_\Gamma \left( Y(A_{1:T}) \prod_{t=t^\star+1}^T \rho_t - f_\theta(H_{t^\star}) \right) \right] =: h(\theta),$$

are proper since they are nonnegative, and finite a.s. at $\theta = 0$. Since the functions

$$\theta \mapsto \frac{\mathbb{1}\{A_{t^\star} = a_{t^\star}\}}{\widehat{\pi}_{t^\star}(a_{t^\star} \mid H_{t^\star})} \ell_\Gamma \left( Y(A_{1:T}) \prod_{t=t^\star+1}^T \widehat{\rho}_t - f_\theta(H_{t^\star}) \right),$$

$$\theta \mapsto \frac{\mathbb{1}\{A_{t^\star} = a_{t^\star}\}}{\pi_{t^\star}(a_{t^\star} \mid H_{t^\star})} \ell_\Gamma \left( Y(A_{1:T}) \prod_{t=t^\star+1}^T \rho_t - f_\theta(H_{t^\star}) \right),$$

are continuous by linearity of $\theta \mapsto f_\theta$, dominated convergence shows continuity of both the empirical loss $\widehat{h}_n(\theta)$ (a.s.) and population loss $h(\theta)$.

Next, we show that the empirical plug-in loss converges pointwise to its population counterpart almost surely. Since $S(0) = \{\theta^\star\}$ by hypothesis, Lemmas 8, 9 will give the final result. Defining the function

$$h_n(\theta) := \mathbb{E}\left[ \frac{\mathbb{1}\{A_{t^\star} = a_{t^\star}\}}{\widehat{\pi}_{t^\star}(a_{t^\star} \mid H_{t^\star})} \ell_\Gamma \left( Y(A_{1:T}) \prod_{t=t^\star+1}^{T} \widehat{\rho}_t - f_\theta(H_{t^\star}) \right) \right],$$

we write

$$|\widehat{h}_n(\theta) - h(\theta)| \leq |h_n(\theta) - h(\theta)| + |\widehat{h}_n(\theta) - h_n(\theta)|,$$

and show that each term in the right hand side converges to 0 almost surely.

To show that the first term goes to zero, note that since $\widehat{\pi}_{t^\star} \to \pi_{t^\star}$ a.s., we have $\pi_{t^\star}(a_{t^\star} \mid H_t) \geq (2C)^{-1}$ a.s. for all $a_{t^\star}$. This gives

$$|h_n(\theta) - h(\theta)| \leq \left| h_n(\theta) - \mathbb{E}\left[ \frac{\mathbb{1}\{A_{t^\star} = a_{t^\star}\}}{\pi_{t^\star}(a_{t^\star} \mid H_{t^\star})} \ell_\Gamma \left( Y(A_{1:T}) \prod_{t=t^\star+1}^{T} \widehat{\rho}_t - f_\theta(H_{t^\star}) \right) \right] \right|$$

$$+ \left| \mathbb{E}\left[ \frac{\mathbb{1}\{A_{t^\star} = a_{t^\star}\}}{\pi_{t^\star}(a_{t^\star} \mid H_{t^\star})} \ell_\Gamma \left( Y(A_{1:T}) \prod_{t=t^\star+1}^{T} \widehat{\rho}_t - f_\theta(H_{t^\star}) \right) \right] - h(\theta) \right|$$

$$\leq \Gamma C^2 \mathbb{E}\left[ |\pi_{t^\star}(a_{t^\star} \mid H_{t^\star}) - \widehat{\pi}_{t^\star}(a_{t^\star} \mid H_{t^\star})| \cdot \left( Y(A_{1:T})^2 (2C)^{2T} + 2|f_\theta(H_{t^\star})|^2 \right) \right.$$

$$\left. + \Gamma C \mathbb{E}\left[ \left( Y(A_{1:T}) 2 (2C)^T + 2|f_\theta(H_{t^\star})| \right) \cdot Y(A_{1:T}) \cdot \left| \prod_{t=t^\star+1}^{T} \widehat{\rho}_t - \prod_{t=t^\star+1}^{T} \rho_t \right| \right],$$

which has an integrable envelope function under our assumptions and conditional on $\mathcal{E}$. By dominated convergence, we have the result since $\widehat{\pi}_t \to \pi_t$ almost surely (and hence $\widehat{\rho}_t \overset{a.s.}{\to} \rho_t$).

To show that the second term converges to zero, we use the following strong law of large numbers for triangular arrays.

**Lemma 10** (Hu et al. [14, Theorem 2]). *Let $\{\xi_{ni}\}_{i=1}^n$ be a triangular array where $X_{n1}, X_{n2}, \ldots$ are independent random variables for any fixed $n$. If there exists $\xi$ such that $|\xi_{ni}| \leq \xi$ and $\mathbb{E}[\xi^2] < \infty$, then $\frac{1}{n} \sum_{i=1}^n (\xi_{ni} - \mathbb{E}[\xi_{ni}]) \overset{a.s.}{\to} 0$.*

The random variable

$$\frac{\mathbb{1}\{A_{t^\star} = a_{t^\star}\}}{\widehat{\pi}_{t^\star}(a_{t^\star} \mid H_{t^\star})} \ell_\Gamma \left( Y(A_{1:T}) \prod_{t=t^\star+1}^{T} \widehat{\rho}_t - f_\theta(H_{t^\star}) \right)$$

are i.i.d. for each trajectory, conditional on $\mathcal{D}_\infty$. By convexity, the below random variable upper bounds the preceeding display

$$\xi = 16\Gamma(2C)^{2T} \left( f_\theta(H_{t^\star})^2 + Y(A_{1:T})^2 \right)$$

on the event $\mathcal{E}$. From hypothesis, we have $\mathbb{E}[\xi^2 \mid \mathcal{D}_\infty, \mathcal{E}] < \infty$. Applying Lemma 10 conditional on $\mathcal{D}_\infty$ and $\mathcal{E}$, we conclude

$$|\widehat{h}_n(\theta) - h_n(\theta)| \overset{a.s.}{\to} 0.$$

Applying Lemmas 8, 9, we conclude that for any $\varepsilon_n \downarrow 0$, $\liminf_{n\to\infty} \text{dist}(\theta^\star, S_{\varepsilon_n}) \overset{p}{\to} 0$ conditional on $\mathcal{D}_\infty$ and $\mathcal{E}$. Now, note that for any $\Delta > 0$,

$$\mathbb{P}\left( |\liminf_{n\to\infty} \text{dist}(\theta^\star, S_{\varepsilon_n})| \geq \Delta \right) = \mathbb{P}\left( |\liminf_{n\to\infty} \text{dist}(\theta^\star, S_{\varepsilon_n})| \geq \Delta \mid \mathcal{E} \right)$$

$$= \mathbb{E}\left[ \mathbb{P}\left( |\liminf_{n\to\infty} \text{dist}(\theta^\star, S_{\varepsilon_n})| \geq \Delta \mid \mathcal{D}_\infty, \mathcal{E} \right) \mid \mathcal{E} \right]$$

$$= \mathbb{E}\left[ \mathbb{P}\left( |\liminf_{n\to\infty} \text{dist}(\theta^\star, S_{\varepsilon_n})| \geq \Delta \mid \mathcal{D}_\infty, \mathcal{E} \right) \right]$$

where the first and the last equality used since $\mathbb{P}(\mathcal{E}) = 1$. By dominated convergence, the preceeding display goes to 0 as $n \to \infty$.

**Figure 2.** Sepsis simulator design sensitivity. Data generation process with level of confounding $\Gamma^\star = 1.0$. Estimated lower and upper bound of two policies (with and without antibiotics) under (a) our approach with sensitivity 1.7 (b) naive approach with sensitivity 1.23.

**Figure 3:** Sepsis simulation with $\Gamma^\star = 2.0$. Dashed lines represent 95% bootstrap quantiles.

# E  Experimental details

This section provides implementation details for the experiments presented in the main text.

## E.1  Management of sepsis patients

We refer the reader to `https://github.com/clinicalml/gumbel-max-scm` for more information on the sepsis simulator developed by Oberst and Sontag [38].

**Simulator**  Oberst and Sontag [38]'s simulator state space consists of a binary indicator for diabetes, and four vital signs {heart rate, blood pressure, oxygen concentration and glucose level} that take values in a subset of {very high, high, normal, low, very low}; size of the state space is $|\mathcal{S}_t| = 1440$. There are three binary treatment options {antibiotics, vasopressors, and mechanical ventilation} ($|\mathcal{A}_t| = 2^3$). In our experiments, simulation continues either until at most $T = 5$ (horizon) time steps, death (reward -1), or discharge (reward +1). Patients are discharged when all vital signs are in the normal range without treatment. Patients die if at least three vitals are out of the normal range. We refer the reader to `https://github.com/clinicalml/gumbel-max-scm` for details regarding the simulator.

**The optimal policy**  Recall that we assume that the decisions are made near-optimally. To learn the optimal policy, we generate 2000 samples for each transition and constructed the transition matrix $P(s, a, s')$ and the reward matrix $R(s, a, s')$ of the MDP. Similar to Oberst and Sontag [38] we used policy iteration to learn the optimal policy. We create a near-optimal (soft optimal) policy by having the policy take a random action with probability 0.05, and the optimal action with probability 0.95. The value function (for the optimal policy) was computed using value iteration. The horizon is $T = 5$ and the discount factor $\gamma = 0.99$, which results in soft optimal policy having an average value (over the possible distribution of state states) of 0.14.

**Confounding**   We injected confounding in the first decision of this simulation by defining two different policies: "with antibiotics" and "without antibiotics". "with antibiotics" which is identical to the soft optimal policy except that the probability mass of actions without antibiotics is moved to the corresponding action with antibiotics. For example, if the probability of the action $a_1 =$(antibiotics on, vasopressors off, ventilation on) in the soft optimal policy is $p_1$, and $a'_1 =$(antibiotics off, vasopressors off, ventilation on) is $p'_1$, then in the "with antibiotics" $a_1$ has probability $p_1 + p'_1$ and $a'_1$ has probability zero in this new policy. The "without antibiotics" does the opposite: moves probability mass of actions with antibiotics to the corresponding action without antibiotics. In our confounding scenario, for healthy patients we administer antibiotics (i.e. follow the "with antibiotics") policy with a higher probability (w.p. $\frac{\sqrt{\Gamma}}{1+\sqrt{\Gamma}}$). For unhealthy patients, we administer antibiotics with a lower probability (w.p. $\frac{1}{1+\sqrt{\Gamma}}$).

Concretely, to compute the transition from a state conditional on an action, we do inverse transform sampling: we generate a uniform random variable $U_t$ on $[0, 1]$, and use this to index into the transition probability distribution for the next state, sorted by the states' value function and current reward. This coupling ensures that if $U_t$ is large, then the next state will have a high value, and if $U_t$ is small, then the next state will have a low value. The hidden variable $U$ used for confounding in the first decision is $U = \sum_{t=1}^{T} U_t$, which serves as a surrogate for the health of patient, because the larger $U$ is, the more likely the patient is to have improving state values. We choose a threshold $u_0$, and if $U > u_0$, the behavior policy follows the action with antibiotics, and if $U \leq u_0$, the behavior policy follows the action without antibiotics, thus introducing confounding.

After the first decision, the behaviour policy is a mixture of two policies: $85\%$ the soft optimal policy and $15\%$ of a sub-optimal policy that is similar to the soft optimal but the vasopressors action is flipped. For example, if probability of the action $a_1 =$(antibiotics on, vasopressors off, ventilation on) is $p_1$, and $a'_1 =$(antibiotics on, vasopressors on, ventilation on) in $p'_1$ in the soft optimal policy, then the sub-optimal has probability $p'_1$ and $p_1$ for action $a_1$ and $a'_1$, respectively.

**Loss minimization**   Since the state and action space are discrete, we learn the tabular value $\kappa(s, a)$ for each state action pair separately to minimize the empirical loss. Additionally, in order to compute the upper bound of both ours and the naive method, we compute the negative of the lower bound on the negative of return (cost).

**Behaviour Policy**   We estimate the behaviour policies from the data in two parts: the first time step and time steps $t = 2$ through $t = 5$. By the assumptions stated above, each of these policies depends only on the previous state, and we learn the tabular probability of each state action pair $\pi_t(a|s)$ separately.

**Design Sensitivity**   We present another design sensitivity experiment, with $\Gamma^\star = 1.0$. Figure 2 (a,b) shows design sensitivity of our method (1.7) versus the naive method (1.23).

**WIS bootstrap**   We showed in Section 2 that in existence of an unobserved confounding OPE estimates that (falsely) assume sequential ignorability will be biased. Figure 3 is the same plot as figure 1 (a) where here we include 95% bootstrap quantiles of weighted importance sampling (WIS) instead of its point estimate. While the bootstrap sampling does not account for the influence of estimating the nuisance parameters (the importance sampling weights) on the evaluation policy value estimate or the bias introduced by using WIS instead of IS, it provides inference conditional on these nuisance parameter estimates, providing a surrogate for the statistical uncertainty. They suggest that quantifying uncertainty in WIS does not fully capture the bias introduced by the unobserved confounder.

### E.2   Autism

In the autism experiments, our data generation process (simulator) is adopted from Lu et al. [31, Appendix B]. Each individual has a set of covariates $X$, consisting of six mean-centered features: {`age, gender, indicator of African American, indicator of Caucasian, indicator of Hispanic, indicator of Asian`}. The Autism SMART trial [23] simulator specifies a set of 300 individuals: to obtain a sample size $N$, we sample with replacement from this set. For details on the simulator, we refer to Appendix B of Lu et al. [31]. At the first timestep there are two actions

available $A_1 \in \{-1, 1\}$, where $A_1 = 1$ denote BLI, and $A_1 = -1$ denote AAC. At the second timestep there are three actions $A_2 \in \{-1, 0, 1\}$, where $A_2 = 1$ denote assigning intensified BLI to slow responders, $A_2 = -1$ denote assigning AAC to slow responder and $A_2 = 0$ denote continuing with the same action for fast responders.

**Confounding**   The original simulator did not have confounding. We now describe how we introduce confounding in this setting.

Lu et al. [31, Appendix B] specifies the effect of the second action (whether to augment BLI with AAC) on the reward outcome $Y$ as follows:

$$Y = \eta_{31}^T X + \eta_{22} Y_0 + \eta_{33}^T A_1 + \eta_{34} Y_{12} - 2\theta(1 - R)(A_1 + 1)A_2 + \epsilon.$$

$A_1$ is either $-1$ or $1$. Therefore the final term (outside of the noise $\epsilon$) is non-zero only when $A_1 = 1$, and we can interpret $\theta$ as the effect size of the adaptive policy (which always takes $A_1 = 1$); for exact definition of the effect size refer to Lu et al. [31]. For those more familiar with the RL literature, it is related to the advantage function. In the original paper, Figure 7 in Lu et al. [31] were generated using 4 different values of $\theta$. The parameters used in these simulations are in the range reported by Lu et al. [31].

We introduce confounding by varying $\theta$ (thereby impacting the potential outcome) and then altering the behavioral treatment decisions according to the knowledge of that $\theta$. More precisely, given a $\theta_0$, for each individual, we randomly set $\theta_0 + \sigma_\theta$ or $\theta_0 - \sigma_\theta$. The second action is 1 with probability $\frac{\sqrt{\Gamma}}{1+\sqrt{\Gamma}}$ if $\theta \geq \theta_0$ and 1 with probability $\frac{1}{1+\sqrt{\Gamma}}$ if $\theta \leq \theta_0$. In our experiments, we take $\sigma_\theta = 5$.

**Loss minimization**   To estimate $\kappa(H_{t^\star}; a_{t^\star})$ in the loss minimization problem, we used a neural network with 3 hidden layers of size {128, 128, 128, 64} with Relu activations, followed by a single linear output layer. We initialize the layers with Xavier initialization and used the Adam optimizer with learning rate $10^{-3}$. The input $H_t$ is 10-dimensional consisting of 6 covariates, indicator of slow responder, initial action $A_1$, number of speech utterances after the initial action, and an interaction term between $A_1$ and the slow responder indicator.

**Behaviour Policy**   We use logistic regression to estimate the behaviour policy from the observed data: note that this is not the true behavior policy, because that depends on the (latent) confounding. Different models were fit for the first and second time steps. For the first timestep the learned model is $\pi_1(A_1|H_1)$, where $H_1$ contains the observed $X$ (6 covariates), and $A_1 \in \{-1, 1\}$. For the estimated behavior policy in the second timestep $\pi_2(A_2|H_2)$, $H_2$ includes $X$ (6 covariates), the action $A_1$, indicator of slow responder, the interaction term between $A_1$ and the indicator, and the number of speech utterances after the initial action.

**Figure 4.** (a) Autism simulation. Outcome of two different policies, confounded adaptive policy (BLI+AAC) and un-confounded non-adaptive policy (AAI). Data generation process with the level of confounding $\Gamma^\star = 2.0$. (b) Autism simulation design sensitivity. Data generation process with the level of confounding $\Gamma^\star = 1.0$. True value of adaptive (BLI+AAC) and non-adaptive (AAC) policies along with estimated lower bound on outcome using our and naive approach

**Extra Experiments**   In the second setting (Case II), the BLI+AAC policy is better than the AAC policy; we again use $\Gamma^\star = 2.0$ to generate data. Standard OPE estimates again overestimate the

outcome for the BLI+AAC policy (Figure 4(b)). The naïve lower bound results in a conservative lower bound that again indicate no conclusions can be drawn about the relative performance of BLI+AAC versus AAC. Our method certifies the superiority of the BLI+AAC policy up to $\Gamma = 4.2$, providing useful certificates under nontrivial levels of confounding.

Figure 4 (c) plots the design sensitivity of our method against the naïve approach (3), when there is in fact no confounding in the data generation process ($\Gamma^\star = 1$). Compared to the naive approach (design sensitivity is $\Gamma = 1.32$), our method allows certifying robustness of the finding—that the adaptive policy is advantageous—up to realistic levels of confounding (design sensitivity is $\Gamma = 2.78$).



[Supplementary Material 2 · sepsis-sensitivity-ours-5.pdf]



**Design Sensitivity, Ours**

- with antibiotics
- without antibiotics

Outcome $\mathbb{E}[Y(\bar{A}_{1:\tau})]$ vs. Level of confounding ($\Gamma$)

[Supplementary Material 3 · sepsis-sensitivity-naive-5.pdf]



Design Sensitivity, Naive

Outcome $\mathbb{E}[Y(\bar{A}_{1:\tau})]$ vs. Level of confounding ($\Gamma$)

- with antibiotics
- without antibiotics