[Reviews · NeurIPS 2020]

Review 1

Summary and Contributions: This paper deals with the problem of Off-Policy Evaluation under Unobserved Confounding. Specifically, it deals with single decision confounding under assumptions on bounds of the "amount of confoundness". They show that a lower bound over the likelihood of potential outcomes can be used to lower bound the expected evaluated return. Then, using convex duality, they give a method of estimating this lower bound. They show consistency of their approximation method in the realizable linear setting. Finally, they illustrate their method on a sepsis simulator as well as a simulator for autistic children.

Strengths: OPE under confounding is an extremely difficult problem. In almost every realistic setting this problem is non-identifiable. It is thus imperative to search for methods that can mitigate the confounding bias through bounds and / or weak assumptions. The main contribution of this work is based on Proposition 1 and Theorem 2 where the authors provide an analytical formulation of the value through the potential outcome likelihood, and a method to lower bound this expression in a scalable manner. The theorems and proofs seem correct. Overall, this work seems relevant and important to researchers in causal inference and OPE.

Weaknesses: The main issue I have is the assumptions made in the paper. I will divide this into two items: grounding and validation. By grounding I mean, how realistic are the assumptions made, and by validation I mean, how can someone test for these assumptions given real data? While I understand that some of the assumptions are model assumptions, I think it would be nice to discuss methods that such model assumptions could be validated. If we cannot validate our models then we have no way of knowing how applicable our assumptions are. Specifically regarding each assumption made in the paper: 1. Assumption A (overlap): this assumption is fine and reasonable in a tabular setting. How would you generalize it to the function approximation setting? Since you do mention this in the section about consistency, I think it would be beneficial to discuss this. 2. Assumption B: This assumption seems reasonable, since mostly we have control over the evaluation policy. 3. Assumption C (sequential ignorability given confounder): This assumption is general and is thus not really an assumption. 4. Assumption D (confounding bound): This is a standard assumption in causal inference. It would be interesting though if you could discuss its relation to the assumption of Zhang and Bareinboim [53] for sequential boundness. 5. Assumption E (single confounder): This is an assumption I am having a hard time with. First, I cannot see a realistic scenario in which this assumption would hold. I can believe that the bound on confoundess may diminish with time, or even be very small, but it is unreasonable to believe that the confounding bias would be zero for all other time steps. Second, with regard to validation, how can one validate this assumption? How can one find t* for which it might hold approximately? What happens when this assumption breaks? 6. Assumption F: (confounding bound): This is again a standard assumption.

Correctness: Yes

Clarity: The paper is easy to follow overall. I feel that Section 4 could be improved -- especially from the point you address Lemma 3 (lines 242 to 276), it felt a little bit rushed and I had to read every sentence twice to make sure I follow. Also, if you are accepted, I think it would be great if you could add a page for discussion, explaining a bit more the limitations of your model and possible future directions.

Relation to Prior Work: This work is related to several recent works on OPE and reinforcement learning with confounding and partial observability. Three more recent works that were not mentioned, and may be beneficial to discuss are: "Warm Starting Bandits with Side Information from Confounded Data": https://arxiv.org/pdf/2002.08405.pdf "Bandits with Partially Observable Offline Data": https://arxiv.org/pdf/2006.06731.pdf "Combining Offline Causal Inference and Online Bandit Learning for Data Driven Decisions": https://arxiv.org/pdf/2001.05699.pdf

Reproducibility: Yes

Additional Feedback: - Aside from the main item of the underlying assumptions, it was not clear to me how loose the lower bound proposed is. Could the authors elaborate? - Lastly, it feels like the extension to non single decision confoundness is not too far. Do the authors believe this is a great step that is out of the scope of their current work? After reading the authors' comments and other reviews I have decided to keep my current score.


Review 2

Summary and Contributions: Consider sequential decision scenarios. This paper introduces a model for estimating how well an evaluation policy had performed, given that we observed a behavioral policy (and its consequences). This is called an off-policy policy evaluation, or OPE. Essentially, this is equivalent to estimating a regret under an arbitrary, counterfactual evaluation policy. The paper argues that this cannot be done in general, because confounders might introduce statistical dependencies that hinder the counterfactual evaluation. To address this, the paper proposes constructing worst-case bounds on the OPE. This is achieved through two main assumptions. First, only a single decision depends on confounders (i.e. the action is independent of the future given the history and the confounding variable). Second, it suggests imposing an upper bound (Gamma) on policy ratios for different values of the confounder, effectively limiting its impact on the future realization. Together, (plus a dual relaxation) these assumptions allow formulating bounds. These depend crucially on the choice of Gamma. Experiments show that these assumptions lead to reasonable estimations.

Strengths: - The paper reads very well (except some of the denser math parts)! It was a pleasure to read. - The problem addressed is relevant - finding a good solution could have a wider impact in real-world applications. - As far as I could see, the arguments are sound and the derivations correct.

Weaknesses: - The presented analysis is limited to a single decision affected by a confounding variable. It is a sensible assumption in many cases, but I got dissapointed by the fact that the current analysis doesn't seem to **conceptually** extend to the general case. - The bounds seem really hard to compute. It would have been nice had the paper provided a (perhaps more conservative) yet simpler expression alongside the current one. - Really, the paper presents a **family of bounds** parameterized by Gamma, but Gamma isn't an easily interpretable quantity. Can this be rephrased in terms of a more standard notion (e.g. an information quantity)?

Correctness: As far as I could tell, the only mistake (?) I could spot is the bar on the history in Assumption A. Otherwise it seems correct.

Clarity: The paper is very well-written and clear. The only thing that wasn't clear to me is how to get the upper bound.

Relation to Prior Work: I was wondering whether this was an incremental contribution in the light of Yadlowski et al.'s prior work.

Reproducibility: Yes

Additional Feedback: POST REBUTTAL: I'd like to thank the authors for their kind clarifications.


Review 3

Summary and Contributions: This paper extends the sensitivity analysis method by Yadlowsky et al. (2018) from ATE estimation in non-sequential setting to off-policy learning in sequential setting in presence of unmeasured confounding. The authors provide interesting examples to illustrate that in sequential setting the confounding bias can be dramatic if decisions in many periods are allowed to be confounded. So they consider the simplified setting where only one of the actions is influenced by unmeasured confounders, where approach of reducing sensitivity bounds computation to ERM still applies. I read other reviews and the author rebuttal, and I'd like to maintain my current rating.

Strengths: The method developed in this paper has solid theoretical ground, and reasonable empirical evaluations. The illustrative toy example that demonstrates how time horizon may aggravate the confounding bias in sequential setting is insightful. This paper contributes to a sensitivity analysis approach to address confounding bias in sequential decision making. This seems like an underexplored but crucial problem, which is particularly relevant for researchers in domains like healthcare where collected data are often observational.

Weaknesses: I think this work is overall pretty good under the framework of sensitivity analysis (which of course is subject to the limitations of sensitivity analysis). I only have some minor suggestions: 1. I like the simple example at the end of section 3 that illustrates the confounding bias when multiple actions can be confounded. I think it'd be good to revisit this challenge after presenting the sensitivity analysis approach, e.g., by pointing out that in this very challenging setting, it is expected that with even relatively small confounding in every period, the sensitivity bounds must be very wide and ultimately not useful. 2. I personally found figure 1(d) to be hard to understand at first glance, partly because some are results are shown in lines while others are in points. Also the points are actually lower bound estimates. The authors may use lines throughout, and also point out that the points represent lower bounds in the caption or legend.

Correctness: Yes

Clarity: Yes

Relation to Prior Work: Yes

Reproducibility: Yes

Additional Feedback:


Review 4

Summary and Contributions: The paper considers evaluating a sequential treatment policy using observational data from another policy where one of the actions in the sequence is confounded by an unobserved factor. The paper extends non-sequential sensitivity analysis by showing how to calculate a bound on the bias created by the unobserved confounder.

Strengths: Sequential decisions are common in healthcare and create ample observational data which can be used to evaluate new policies. But unobserved confounding is common. As the presence of unobserved confounding cannot be determined, it is necessary (and standard practice) to establish sensitivity bounds based on hypothetical confounders, which is done here. The authors give plausible examples where only one of the actions may have an unmeasured confounder, establishing relevance. It appears that this setting has not been considered before, establishing novelty. The main theoretical analysis is strong and clearly explained. Empirical evaluation is done on two synthetic simulators mimicking real-world scenarios.

Weaknesses: UPDATE: After reading the author response, it seems that my concerns about practical relevance are real, but there exist situations where the concerns don't apply. Additionally, the authors clarified that their method is fairly easy to implement and to use with non-linear models. I am updating to a weak accept. Though the theoretical results appear sound (but not scientifically surprising), a key selling point is the practical relevance to clinical settings. Some concerns about practical usefulness/relevance come up: 1) The overlap assumption seems problematic in the settings of interest, e.g. healthcare workers following protocol. If protocol is followed, actions are deterministic, violating overlap. If not, it seems likely that unobserved confounding is not limited to one action. This may limit the method’s use cases. 2) Sequential ignorability for the evaluation policy: if e.g. unobserved comorbidities are used to inform decisions, it seems clinically questionable to propose and evaluate a new policy that ignores them? 3) Though examples are provided, I cannot assess without clinical knowledge how plausible it is in practice that unobserved confounding is limited to one action (or a small proportion of actions). As far as I can see, no references are given. 4) The empirical approximation seems complicated and relies on a linearity assumption. I expect that applied studies will avoid it for these reasons. However, I do not understand the approximation well enough to evaluate it fully. A simpler method, e.g. based on Monte-Carlo techniques, might be helpful in this regard, and computationally feasible as T is likely to be small in practice. My evaluation is therefore uncertain, depending especially on concerns 1-3.

Correctness: Yes.

Clarity: The paper is well-written and easy to digest up to L236-298. This part could benefit from more reader-friendly exposition or shortening to make more room for experiments.

Relation to Prior Work: Yes this is very clear.

Reproducibility: Yes

Additional Feedback: Typo: “as the reward/outcome”

[Author Response · NeurIPS 2020]

Thanks to the reviewers for the helpful comments. We'll address them all in the paper & answer the key questions here.

**Single-decision confounding (R1, R4)** We completely agree that multi-decision confounding is also important, but
note our model is already more realistic than sequential ignorability, which is assumed in almost all existing work on
OPE. Under our model, an unobserved confounder can affect rewards or state transitions in other time steps, but only
directly affect the action in a single time step. Since the state transition and action taken reveals new information, it is
sometimes reasonable to assume that an unobserved confounder in one time step is implicitly observed in the next. We
provide several sample scenarios where single decision of confounding is reasonable, including evaluation of automated
sepsis treatment policies. If OPE is robust to confounding at multiple times, it should be robust to confounding in any
one time, which we can check with our method. Therefore, our method provides nontrivial necessary conditions for
reliable OPE estimates, and is a step towards ensuring more robust OPE, as we will clarify in the text.

**Testing assumptions (R1)** Testing assumptions is important, and some of the standard assumptions we make such as
overlap can be tested statistically. However, sequential ignorability (SI) cannot be tested from off-policy data, and our
method precisely allows analyzing sensitivity to violations to this untestable assumption. As such, the relaxations of SI
we consider (bounded confounding in Assumption D, E) can be thought of as the extent to which SI is violated.

**Looseness (R1)**: Our bound is tight when the final decision is confounded; assessing conditions for tightness in earlier
timesteps appears challenging, and is left to future work. We will add this discussion in the final version.

**Assump. D and relation to Z & B 2019 (R1)** We briefly discussed this in Lines 76-80 and will highlight it further.

**Flexibility of loss minimization and simplifying computation (R2, R4)**: While our notation is cumbersome due to
the complexity of the sequential setting, our procedure is ultimately a loss minimization problem with importance
sampling (IS) weights. We can minimize the loss using any model class (e.g. deep nets; not just linear models), with
standard ML training algorithms. We only use the linearity assumption in our proof to guarantee statistical consistency.
Consistency would hold with many other ML models, but we use a linear model for simpler exposition. The naïve
bound in the supplement are simpler to compute, but are often too conservative.

**Extending to multi-step confounding (R1, R2).** When multiple decisions are confounded, we can still write down an
optimization problem over likelihood ratios corresponding to each confounded decision. However, both the objective
and equality constraints are nonconvex, and the problem is over infinite dimensional likelihood ratios. Our approach of
exploiting convex duality to derive the loss minimization problem does not generalize to the multi-step confounding
case. Tractable approximations in the general case is a interesting future direction, which we will discuss in more detail.

**Interpreting $\Gamma$ (R2)**: As we discuss in lines 190-192, for binary actions, the confounding model we study has a crisp
interpretation: $\log \Gamma$ bounds the difference in log odds for two individuals with identical observed states, but different
unobserved confounders. Our model is a natural extension of this to multi-action, sequential settings. In practice, a
meaningful assessment of the robustness of OPE is the threshold level of $\Gamma$ at which the findings of OPE no longer
holds (e.g. bound on evaluation policy's reward become worse than that of benchmark's); we will emphasize this more.

**Relation to Yadlowsky et al. (R2)**: Addressing a sequence of actions requires considering their dependence, which the
methods in Yadlowsky et al. don't do; See lines 86-8, 267-9. Solving these directly enables new multi-step applications.

**Overlap and guidlines/protocols (R1, R4)**: As long as the action space is finite, overlap is a standard assumption,
even in continuous state spaces; continuous actions are beyond our paper's scope (we will clarify this). Indeed, overlap
is a requirement for any OPE algorithm that does not make structural modeling assumptions. It is an intuitive condition,
because lack of overlap implies that there are no trajectories in the data that follow the actions we wish to evaluate.
Note that any stochasticity that only affect the rewards through the decision maker's actions creates overlap.

Guidelines are usually based on strong evidence, and so safe policies to be evaluated should follow these guidelines as
well, again implying overlap. A common source of random variation occurs when clinicians make different decisions
within the guidelines. Important opportunities for policy improvement are where there are large variations in clinical
practices, e.g. guidelines that defer to practitioners, or lack thereof. We are happy to clarify these points in the main text.

**Seq. ignorable eval policy (R4)**: The reviewer's point is a good one, e.g. for evaluation of updated clinical recommen-
dations. However, we focus on automated policies that can only use recorded inputs due to the importance of making
credible claims about AI systems. Even when influencing human decision makers, it is problematic in some cases to
recommend they continue using unobserved variables to influence decision; e.g. when clinicians may be biased or
influenced by spurious correlations. Also, the distribution of confounding variables may also shift over time, between
populations and providers; using only observed variables could provide more robust recommendations.

**Related work and clarity of writing (R1)**: We agree that Section 4 was very dense. In the final version, we will
expand discussion of theoretical results, implications of our model, and connection to other methods with the additional
space. Also, we will include discussion of the suggested references on bandits in our related work.

[Meta-Review · NeurIPS 2020]

The reviewers appreciated the clear writing (with a request to clear up section 4) and theoretical results of this work. There were a number of clarifications requested by reviewers and if the authors can update the paper as they described in the rebuttal it will make for a great contribution to the conference. I vote to accept.